# Population Dynamics of Intestinal *Enterococcus* Modulate *Galleria mellonella* Metamorphosis

Hyun Gi Kong,[a,c] Jin-Soo Son,[a] Joon-Hui Chung,[a] Soohyun Lee,[a] Jun-Seob Kim,[b] Choong-Min Ryu[a]

[a]Infection Disease Research Center, KRIBB, Daejeon, South Korea
[b]Department of Nano-Bioengineering, Incheon National University, Incheon, South Korea
[c]Department of Plant Medicine, Chungbuk National University, Cheongju, South Korea

**ABSTRACT** Microbes found in the digestive tracts of insects are known to play an important role in their host's behavior. Although Lepidoptera is one of the most varied insect orders, the link between microbial symbiosis and host development is still poorly understood. In particular, little is known about the role of gut bacteria in metamorphosis. Here, we explored gut microbial biodiversity throughout the life cycle of *Galleria mellonella*, using amplicon pyrosequencing with the V1 to V3 regions, and found that *Enterococcus* spp. were abundant in larvae, while *Enterobacter* spp. were predominant in pupae. Interestingly, eradication of *Enterococcus* spp. from the digestive system accelerated the larval-to-pupal transition. Furthermore, host transcriptome analysis demonstrated that immune response genes were upregulated in pupae, whereas hormone genes were upregulated in larvae. In particular, regulation of antimicrobial peptide production in the host gut correlated with developmental stage. Certain antimicrobial peptides inhibited the growth of *Enterococcus innesii*, a dominant bacterial species in the gut of *G. mellonella* larvae. Our study highlights the importance of gut microbiota dynamics on metamorphosis as a consequence of the active secretion of antimicrobial peptides in the *G. mellonella* gut.

**IMPORTANCE** First, we demonstrated that the presence of *Enterococcus* spp. is a driving force for insect metamorphosis. RNA sequencing and peptide production subsequently revealed that antimicrobial peptides targeted against microorganisms in the gut of *Galleria mellonella* (wax moth) did not kill *Enterobacteria* species, but did kill *Enterococcus* species, when the moth was at a certain stage of growth, and this promoted moth pupation.

**KEYWORDS** microbiota alteration, insect metamorphosis, host-microbe interactions

**N**ext-generation sequencing (NGS) technologies have significantly contributed to the study of microbiome composition and interactions in insects. Symbiotic partners, in particular, have been linked to host feeding and immune system in some insects (1–3). However, microbe and host interactions in Lepidoptera have received less attention. Consequently, the possible effects of specific microorganisms on Lepidoptera ecosystems, evolution, and development are largely unknown. Lepidopterans, which include butterflies and moths, comprise around 180,000 species, classified into 46 superfamilies, and are among the most numerous and diverse taxa on the planet.

Lepidoptera are holometabolous insects that develop through four stages: egg, larva, pupa, and adult. Each developmental stage is genetically and ecologically distinct. Through metamorphosis, particularly between larval and pupal development, holometabolous insects rebuild their digestive tracts anatomically (4). Reorganization of the host's digestive tract also changes the host microbiota (5). Recent studies have identified alterations in the composition of gut symbionts of Coleoptera, Diptera, Lepidoptera, and Hymenoptera during metamorphosis (6–9). However, the impact of specific microbial symbionts on host

Address correspondence to Jun-Seob Kim, junkim@inu.ac.kr, or Choong-Min Ryu, cmryu@kribb.re.kr.

The authors declare no conflict of interest.

functions during metamorphosis is still not clear (10). It has been proposed that the composition of the insect gut microbiota is shaped by either (i) competition between gut bacteria or (ii) the influence of the host immune system (11). Despite the continued interest in insect immunity, no definitive evidence linking host immunology to bacterial competition throughout the transformation process has been discovered.

*Galleria mellonella*, a member of the Lepidoptera, is known as a parasitic pest of bees and their hives (12). Additionally, *G. mellonella* is a useful animal model system for biological investigations of human-pathogenic microorganisms and for the study of microbial mutual interactions (13). The insect innate immune system is a critical research topic due to its being the simplest model for human immune defense against microbial attacks. A previous study demonstrated that hormones and the immune system of the host may be intimately linked to changes in the gut microbial flora (14). It is well established that 20-hydroxyecdysone, a metamorphosis hormone, modulates production of antimicrobial peptides (AMPs) (15). AMPs involved in the defense against harmful microbial invasion are produced by adipose bodies, hemocytes, salivary glands, and intestinal cells (16). At least 18 antimicrobial peptides are synthesized by intestinal cells, and variations in AMPs during insect development are anticipated to have an effect on the intestinal microbial population of *G. mellonella* (17).

Here, we investigated gut microbiota by conducting 16S rRNA amplicon sequencing and performed transcriptome analysis of *G. mellonella* developmental stages, including eggs, larvae, pupae, and adults, to examine the role of bacterial communities on insect metamorphosis. We established that the structure of microbial communities fluctuates in response to immunological and morphological changes in the host. Our discovery provides the first demonstration of a feedback loop mechanism, through which *G. mellonella* modifies its own gut microflora to initiate metamorphic changes, and establishes the groundwork for future studies on the sophisticated modulation between host immune responses and population dynamics of gut symbionts during the different developmental stages of the host.

## RESULTS

**Alteration of microbiota through the life stages of *Galleria mellonella*.** *G. mellonella* is a moth species that undergoes complete metamorphosis, displaying distinct morphological and functional characteristics at each stage. For example, adults exhibit no feeding behavior, whereas worms (larva) infiltrate honeycombs, consume wax, and feed on bee larvae (18–20). Hence, we hypothesized that there are likely to be changes in the composition of gut microbial species between larvae and adults. Microbiome analysis, using 16S rRNA amplicon sequencing, was performed on each stage (eggs, larvae, pupae, and adults) to investigate possible shifts in gut microbiota. The egg-stage microbial communities contained the greatest number of operational taxonomic units (OTUs) (5,937) showing species-level diversity, and the fraction of species belonging to the genus *Enterobacter* was ∼50% (Fig. 1A). Species diversity was lowest in the larval stage, with only 50 OTUs, the majority of which were species of the genus *Enterococcus*. No substantial alterations were seen between 3rd and 4th instar larvae (Fig. 1A). Surprisingly, the composition of the gut microbiota switched substantially, from *Enterococcus* to members of the genus *Enterobacter* (90%), during the pupa stage, implying that pupation involved some, as yet undiscovered mechanisms that influence the composition of microbial communities (Fig. 1A). The adult stage revealed a 20% increase in the abundance of *Enterococcus* species (Fig. 1A). The indices of evenness (Shannon and Simpson) and richness (ACE and Chao1) of the microbial community for each occurrence of *G. mellonella* were high in eggs, pupae, and adults; however, evenness and richness were very low in the 3rd and 4th instars (Table 1). These results suggest that the intestinal microbiome of *G. mellonella* differs according to development stage, and in particular, *Enterococcus* spp., which account for more than 90% of the larval intestinal microbiome during pupation, are replaced by *Enterobacter* spp. in the pupal stage.

By comparing the gut microbiota of eggs, larvae, and pupae of *G. mellonella*, taxonomic differences relating to each developmental state were detected using the linear discriminant

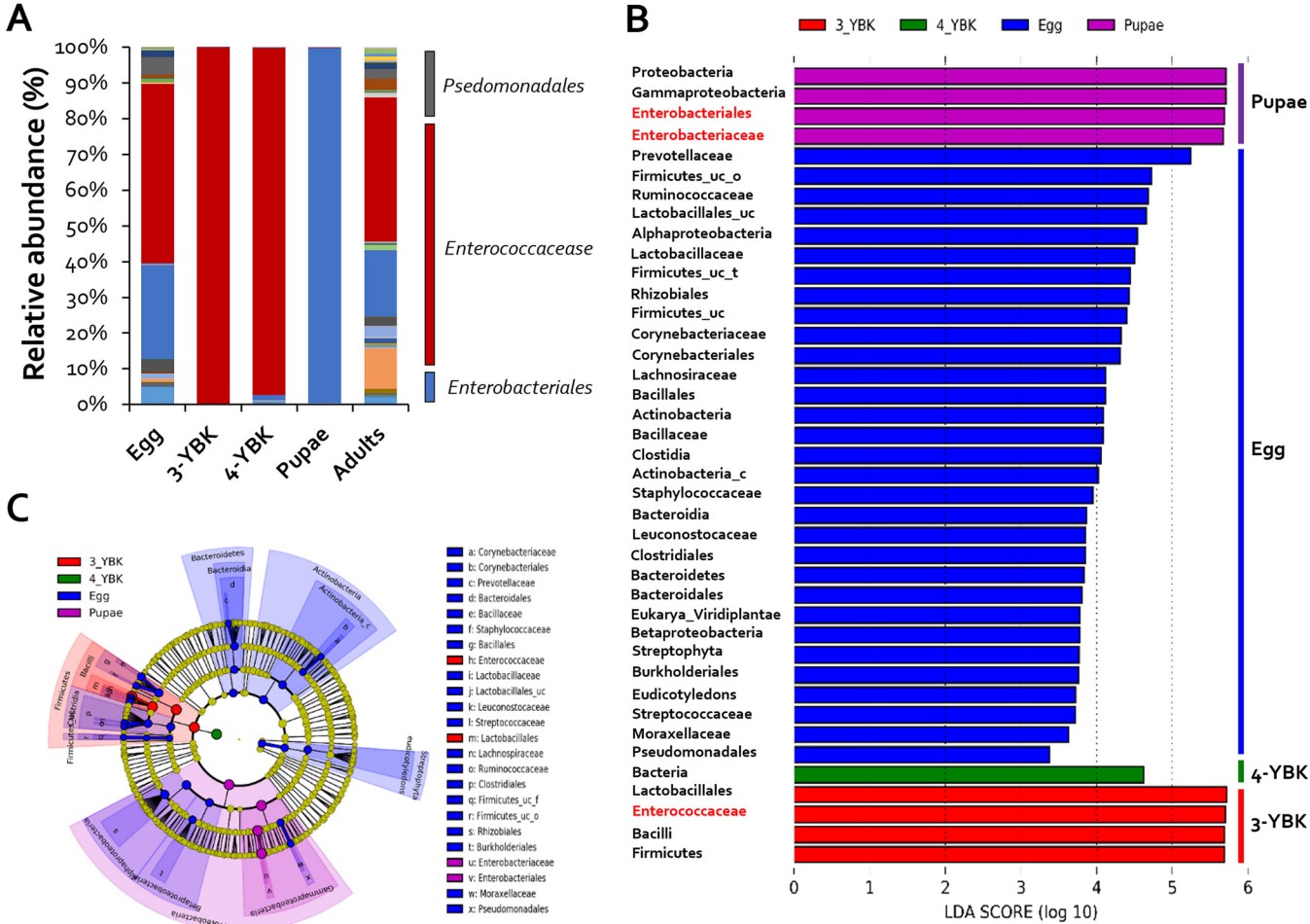

**FIG 1** Changes in symbiont diversity across life stages of *G. mellonella*. (A) Relative abundance (percentage) of bacterial genera at each developmental stage (*n* = 8). (B) A histogram of the linear discriminant analysis scores reveals the most differentially abundant taxa among different development stages: green represents 3rd instar larvae, red represents 4th instar larvae, blue represents eggs, and purple represents pupa-enriched taxa. (C) Cladogram generated by the LEfSe method indicating the phylogenetic distribution of gut microbiota in *G. mellonella* across the development stages from 500 exploratory bootstrap trials. Bar, 0.01 changes per site. The 3rd instar larvae are marked with 3-YBK and 4th instar larvae with 3-YBK.

analysis (LDA) effect size (LEfSe) method (21). LDA scores revealed that *Prevotellaceae* were enriched in eggs, but *Lactobacillales*, *Enterococcaceae*, and *Firmicutes* were enriched in larvae (Fig. 1B and C). On the other hand, *Proteobacteria*, *Gammaproteobacteria*, *Enterobacteriales*, and *Enterobacteriaceae* were abundant during the pupa stage. This demonstrated that species richness is largely division specific in some developmental stages (e.g., Gram positive in larvae and Gram negative in pupae). At the species level, *Enterococcus innesii*, *Enterobacter xiangfangensis*, and *Providencia vermicola* were most abundant in the larval, pupal, and adult stages, respectively. *E. innesii* had the maximum number of reads (12,240) during the larval

**TABLE 1** Sequencing details and $\alpha$ diversity

| Sample | No. of valid reads | No. of OTUs | $\alpha$ diversity index value | | | |
|---|---|---|---|---|---|---|
| | | | ACE | Chao1 | Shannon | Simpson |
| Eggs | 19,122 | 5,937 | 76,520 | 38,466 | 6 | 0.013 |
| Pupae | 19,962 | 2,820 | 50,708 | 25,075 | 3 | 0.149 |
| Adults | 19,607 | 2,981 | 46,866 | 24,810 | 3 | 0.182 |
| | | | | | | |
| Larvae | | | | | | |
| 3rd instar | 19,982 | 47 | 315 | 173 | 1 | 0.468 |
| 4th instar | 19,930 | 49 | 81 | 63 | 1 | 0.400 |

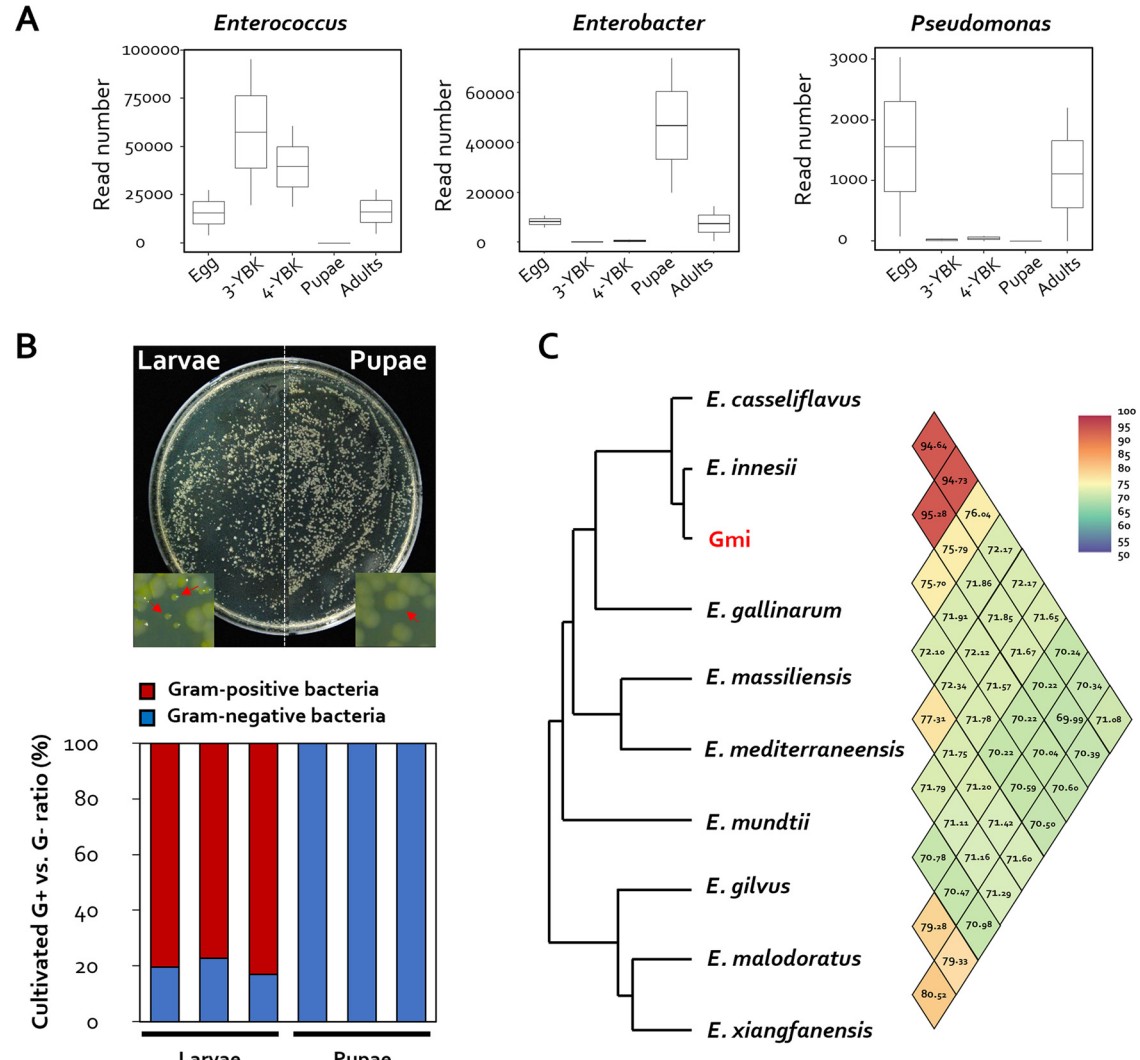

**FIG 2** Isolation of symbionts from *G. mellonella* larvae and pupae. (A) Read number of OTU corresponding to *Enterococcus*, *Enterobacter*, and *Providencia* at each developmental stage (*n* = 12). (B) Investigation of the distribution of bacteria cultured from *G. mellonella* larvae and pupae. Representative bacterial colonies obtained after plating on TSA medium are shown. (C) The average nucleotide identity (ANI) value of the entire genome sequence of the genus *Enterococcus* sp. and the whole-genome information of the isolate were determined using the NCBI database (https://www.ncbi.nlm.nih.gov/).

stage, while *E. xiangfangensis* had the highest number of reads during the pupal stage (12,591). Finally, the prevalence of *P. vermicola* increased to 12,603 reads in adults from less than 10 reads in all other stages (Fig. 2A).

**Isolation of *E. innesii* present in larvae.** Given that investigation of the microbiota of *G. mellonella*, based on developmental stages, revealed a distinct shift in microbial composition, we hypothesized that keystone taxa might be associated with specific *G. mellonella* developmental stages. The larval and pupal phases, in which significant shifts in the microbiota had been demonstrated, were the target stages investigated. To investigate keystone taxa (i.e., the predominant bacterial species) present during the pupation process (from larva to pupa), isolates from two candidate genera, *Enterococcus* (Gram positive) and *Enterobacter* (Gram negative), were cultured from each larval and pupal stage, respectively. Total cell counts of cultivated bacteria in larvae and pupae were comparable: $2.8 \times 10^7$ ($\pm$76.22) and $2.7 \times 10^7$ ($\pm$231.32) CFU, respectively. The diversity and makeup of microbial communities, on the other hand, varied greatly. Colonies with two distinct morphologies were detected in larvae, while colonies with only a single distinct phenotype were identified in pupae (Fig. 2B). Species classification was performed for Gram-positive bacteria,

which are dominant in *G. mellonella* larvae. Whole-genome analysis had previously identified an *Enterococcus* species in the gut of *G. mellonella* (22), and based on relevant whole-genome data, here we identified this species as *Enterococcus innesii* (Fig. 2C).

**Role of intestinal *Enterococcus* in *G. mellonella* pupation.** To corroborate the role of intestinal *Enterococcus* spp. during the pupation stage, *G. mellonella* larvae were infected with rifampin-resistant (Rif$^r$) *E. innesii* and Rif$^r$ *Enterococcus mundtii* and then treated with 100 $\mu$g/mL vancomycin, an antibiotic that targets only Gram-positive bacteria (Fig. 3A). After vancomycin treatment, the presence of Rif$^r$ *E. innesii* and Rif$^r$ *E. mundtii* in larval intestines was evaluated by culturing isolated bacteria on rifampin-containing medium. As a result, it was confirmed that each strain was present in the intestine at $10^5$ to $10^6$ CFU/g 24 h after feeding. However, when treated with vancomycin, both strains were present at less than 10 to 100 CFU/g (see Fig. S4 in the supplemental material), indicating the effectiveness of vancomycin. Following antibiotic challenge, pupation rates in the absence of feeding and at 30℃ were assessed. The pupation rate of larvae treated with polymyxin B, an antibiotic that targets only Gram-negative bacteria, was the same as that of larvae treated with phosphate-buffered saline (PBS) (Fig. 3B). However, the pupation ratio of the larvae treated with the vancomycin was 80% on the 3rd day, which was 4 times higher than that of the PBS treatment, which showed a pupation rate of 20% (Fig. 3C). In addition, by the 5th day after treatment, there was 100% pupation rate in the presence of vancomycin, whereas only 20% of the larvae in the PBS control group pupated (Fig. 3C). This finding led us to speculate about a link between pupation and intestinal *Enterococcus* absence or reduction. To substantiate this finding, we evaluated the pupation rates of larvae challenged with PBS administered with *E. innesii* as a control. Treatment of *G. mellonella* with PBS-*E. innesii* resulted in less than 20% pupation by the 5th day, and pupation was later than that in the control group, which exhibited 40% pupation (Fig. 3D).

Treatment with *Enterococcus mundtii*, another *Enterococcus* species commonly detected in the *G. mellonella* intestinal microflora, resulted in a pupation rate similar to that of the PBS-treated group. However, treatment with *Bacillus subtilis* 168, which is not found in the *G. mellonella* intestine, produced similar pupation rates to vancomycin treatment alone. On the other hand, in the case of *E. mundtii*, pupation was recovered to a level similar to that of the antibiotic-free control group as in *E. innesii* (Fig. 3E and F). Taken together, these findings indicate that loss of intestinal *Enterococcus* spp. plays a pivotal role in the pupation of *G. mellonella*.

**Analysis of factors related to pupation of *Enterococcus* bacteria.** When *E. innesii* was additionally administered to vancomycin-treated *G. mellonella*, the pupation rate was restored to the level of the untreated control group. To confirm which factors of *E. innesii* affect the pupation of *G. mellonella*, after culture of *E. innesii* in tryptic soy broth (TSB), the supernatant (extracellular components [ECs]) and bacterial pellet were divided into supernatant (ECs), treated with vancomycin, and fed 2 days later to investigate pupation (Fig. 4A). Pupation in the presence of ECs was identical to that of vancomycin treatment, whereas the pupation rate in the presence of bacterial pellets was similar to that of the PBS control (Fig. 4B). In the TSB treatment group, pupation seems to recover on the 5th day, which indicates the possibility of an effect caused by reculture of the *Enterococcus* remaining in the intestine.

To identify the cellular fraction responsible for inhibition of pupation, intracellular components (ICs) and cell debris fractions were prepared from bacterial cells and tested in the pupation assay: the presence of ICs resulted in similar pupation rates to those seen with vancomycin treatment, whereas the presence of cell debris resulted in lower pupation rates than those seen with PBS treatment (Fig. 4C).

**Transcriptome analysis of *G. mellonella* developmental stages.** Although the possibility of a link between pupation and a reduction in intestinal *Enterococcus* spp. has been suggested, the mechanism by which the host triggers this reduction remains unknown. To investigate this mechanism, host transcriptome analysis of each embryonic stage was undertaken. The overall pattern of gene expression identified differentially expressed genes (DEGs) in larvae and pupae (Fig. 5; Fig. S1). Specifically, 669 and

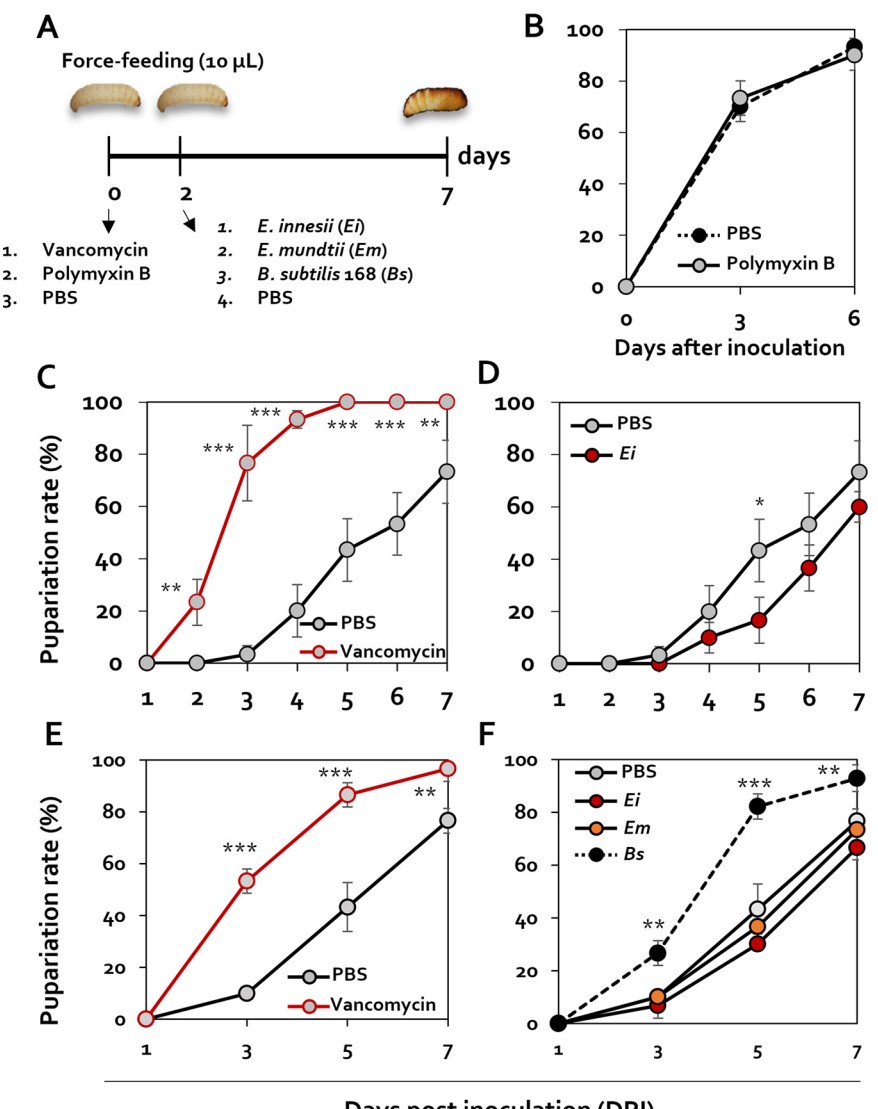

**FIG 3** Changes in metamorphosis in the presence of *Enterococcus innesii*. (A) Schematic diagram showing the regimen for treatment of *Galleria mellonella* larvae with antibiotics and *E. innesii*. (B) Pupation changes after polymyxin B (100 µg/mL) treatments were confirmed using 4th instar larvae. Pupation rates of larva challenged with PBS (black circles on black dotted line) or polymyxin B (gray circles on black line) are shown. (C) The pupal changes of *E. innesii* ($1 \times 10^5$ CFU/larva) treatment and vancomycin (100 µg/mL) treatment were confirmed using 4th instar larvae. Antibiotics and bacteria were fed to the larvae through the mouth, and the degree of pupation was measured by culturing at 30°C for 8 days after feeding. To analyze the recovery of metamorphosis caused by *E. innesii* after antibiotic treatment, a bacterial suspension was fed to larvae 2 days after the antibiotic feeding. Treatment groups were treated with vancomycin (gray circles on black line), control (gray circles on black line), and *E. innesii* treatment after vancomycin treatment (gray circles on red line). (D) Analysis of changes in pupation by *E. innesii* treatment after vancomycin treatment. Shown are results from treatment with *E. innesii* (e.g., black circles on black line) and the control (gray circles on black line). (E and F) Analysis of changes in pupation by *E. innesii* (Ei), *E. mundtii* (Em), and *Bacillus subtilis* (Bs) treatment. All experiments were performed in triplicate with 30 larvae per replicate.

755 genes (false-discovery rate [FDR], 0.05; fold enrichment, >2) were identified as DEGs in larvae and pupa, respectively (Fig. 5A).

Gene Ontology (GO) analysis of DEGs in pupal and larval stages was performed using PANTHER GO-Slim (Fig. 5B to D). In relation to specific molecular functions, the gene group showing hormone activity (GO:0005179) was upregulated (15.14-fold) in the larval stage, whereas gene groups related to cytoskeletal protein binding (GO:0008092) and the structure formation of the cytoskeleton (GO:0005200) were upregulated 3.28- and 3-fold,

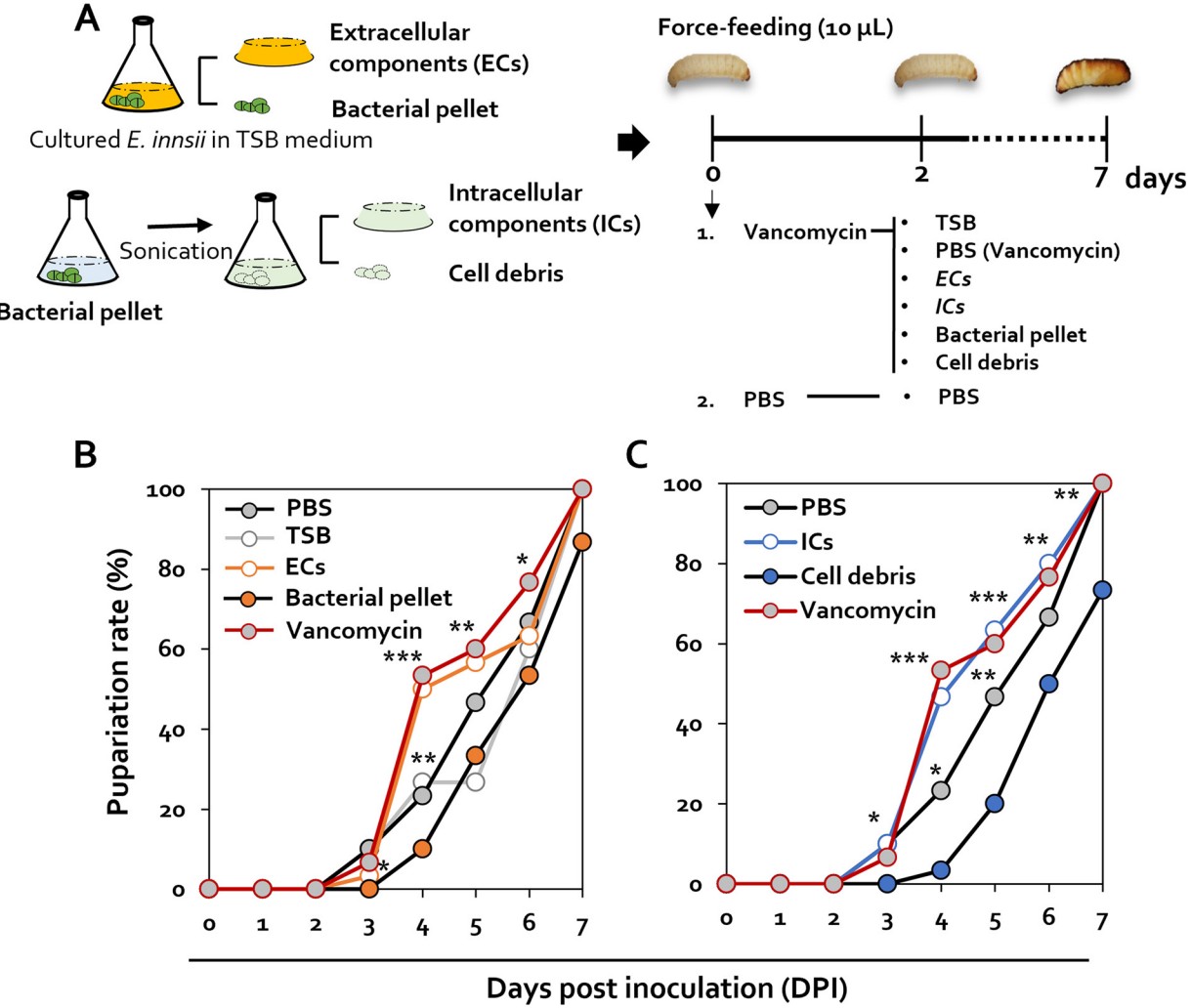

**FIG 4** Pupation changes associated with specific *Enterococcus* cell components. (A) Schematic diagram showing the methods used for isolating the components of *E. innsii* used in the treatment of *G. mellonella* larvae. (B) Pupation changes after treatments with extracellular components (ECs), TSB, PBS, bacterial pellet, and vancomycin were confirmed using 4th instar larvae. (C) Pupation changes after treatments with intracellular components (ICs), PBS, cell debris, and vancomycin were confirmed using 4th instar larvae. All experiments were performed in triplicate with 10 larvae per replicate.

respectively, in pupae (Fig. 4B). These results confirm that, in terms of molecular function, physiological activity and signal transduction are actively performed in larvae, whereas epidermal reorganization is activated in pupae. In the case of cell components, the cell membrane signaling-associated, heterotrimeric G-protein gene (GO:0005834) was elevated (6.31-fold) in the larval stage, compared with the pupal stage, and the microtubule protein genes (GO:0005874 and GO:0005815) were 4.56- and 4.27-fold upregulated, respectively, in pupae (Fig. 5C). Finally, in the case of biological processes, gene groups relating to neuromuscular synaptic transmission (GO:0007274) and cell signaling-associated, cellular calcium ion homeostasis (GO:0006874) were 5.76- and 5.26-fold upregulated, respectively, in larvae compared with pupae, whereas DNA recombination genes (GO:0006310) and immune response-related genes (GO:0006955) were upregulated (7.88- and 5.32-fold, respectively), in pupae relative to larvae (Fig. 5D). In the case of microtubules (GO0005874) and DNA recombination (GO:0006310), it was confirmed that their expression increased significantly in pupae according to the results of sparse canonical correlation analysis (CCA) (Fig. S2). As a result of the Gene Ontology (GO) analysis of the DEGs in the pupal stage, it was confirmed that tissue reorganization proceeded during pupation, and the expression of antimicrobial peptides (AMPs) related to immunity was induced. Our results suggest, therefore,

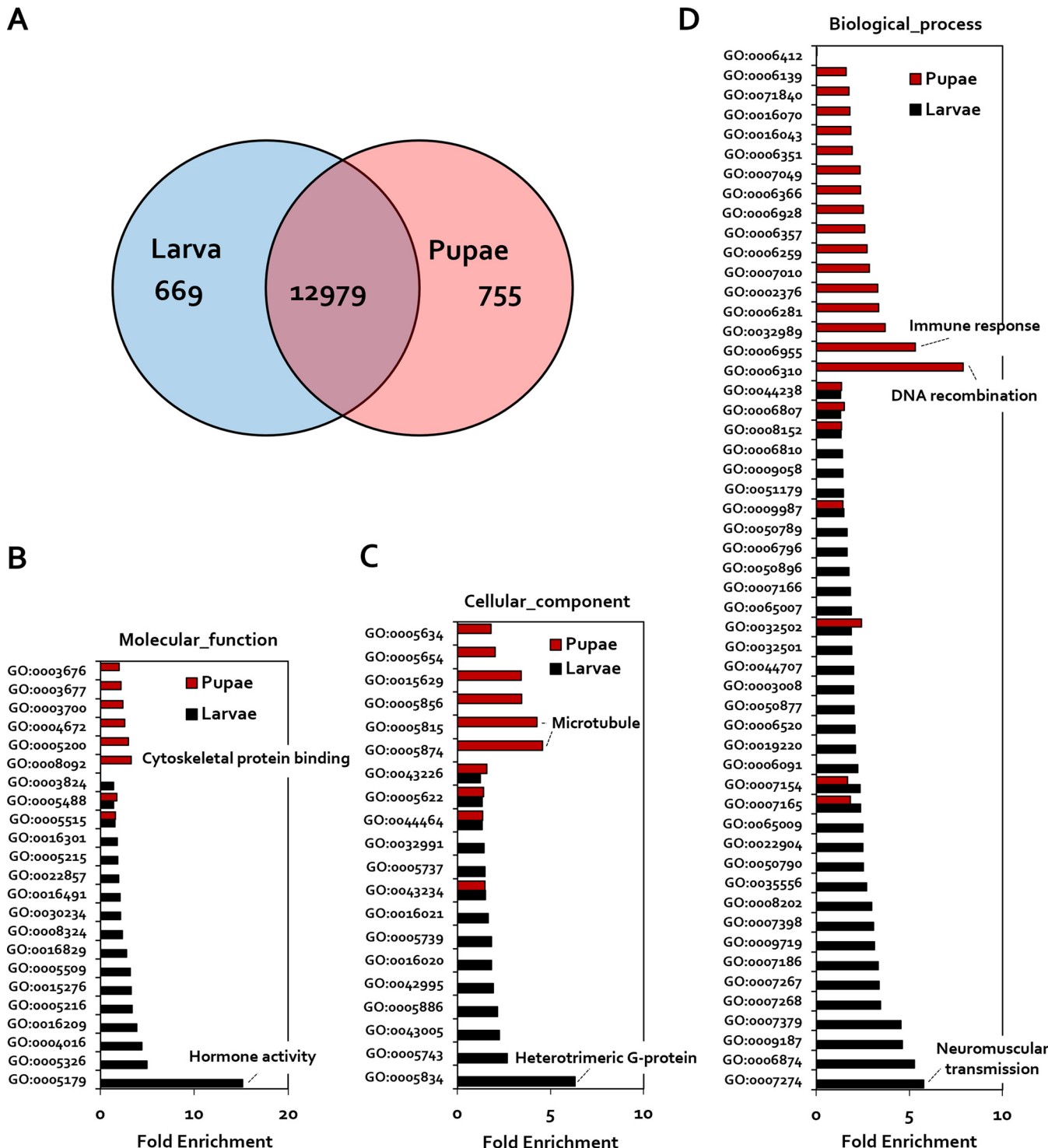

**FIG 5** Profiling of differentially expressed genes and Gene Ontology clustering in the *G. mellonella* following developmental stage. (A) Venn diagram showing differentially expressed genes with the smallest false-discovery rate (FDR): >2. (B to D) Gene Ontology term classification of genes upregulated in larvae or pupae related to molecular function (B), cellular components (C), and biological process (D). The hormone activity (GO:0005179) and immune response (GO:0006955) were significant changes of the transcriptomic pattern between larvae and pupae. Enriched transcripts of pupae are shown in red. Enriched transcripts of larvae are shown in black.

that immune response changes and the expression of AMPs correlate with changes in the intestinal microflora.

**Analysis of the nature and activity of AMPs produced by *G. mellonella* during metamorphosis.** Intestinal cells of mammals and insects produce a variety of AMPs, which function as inhibitors of microorganisms invading the intestine (23). Additionally, it has been

observed that 20-hydroxyecdysone, a metamorphic hormone, regulates the expression of AMPs in *Bombyx mori* (24). As a result, variations in AMPs during different developmental phases may result in alterations in the gut microbial ecology. To confirm the production of AMPs during the larval and pupal stages, the gene expression levels of important AMP genes were evaluated using real-time quantitative PCR (RT-qPCR). HCM000600 and HCM009440 (cecropin A) had an expression value of $4.58 \pm 0.28$ at the larval stage, higher than the levels of prepupal ($0.53 \pm 0.04$) and pupal ($1.67 \pm 0.09$) expression. On the other hand, levels of expression of HCM007113 (cecropin D), HCM008333 (gallerimycin), and HCM003020 (LYS) were $4.48 \pm 1.09$, $2.30 \pm 0.46$, and $4.59 \pm 0.31$ in the pupal stage, respectively, which was higher than their levels of expression in the larval stage ($2.54 \pm 0.42$, $0.63 \pm 0.11$, and $0.61 \pm 0.52$, respectively) (Fig. 6A).

Quantitative analysis of AMPs by SDS-PAGE was performed on samples obtained from different developmental stages in the metamorphosis of *G. mellonella*. Proteins were extracted from 200 mg each of larvae, prepupae, and pupae. For the analysis of AMPs, proteins of less than 25 kDa were separated by in-gel chymotryptic digestion. Protein profile analysis of peptide mixture samples obtained by in-gel chymotryptic digestion was performed using liquid chromatography-electrospray ionization quadrupole time of flight mass spectrometry (LC-ESI-Q-TOF-MS) (Fig. 6B; Fig. S3). The peptide sequences obtained following LC-ESI-Q-TOF-MS were qualitatively analyzed using sequence databases of Ditrysia and Lepidoptera. For larvae, a total of 16 detected peptides were analyzed. Apolipophorin-3, known as an antimicrobial protein, was detected in prepupae and pupae, and acanthoscurrin-2-like protein was specifically identified in pupae (Fig. 6B). In conclusion, AMPs such as apolipophorin-3 and acanthoscurrin-2 were identified only in pupae (Fig. 6A and B), suggesting that these AMPs are produced when *G. mellonella* pupates and may alter the intestinal microbiota of the host.

To investigate whether AMPs produced during the pupal stage of the development of *G. mellonella* influence the predominant intestinal bacteria present at each stage, we examined the sensitivity of *E. innesii* (the predominant bacterial isolate in larvae), *E. xiangfangensis*, and *Pseudomonas azotoformans* (the predominant bacterial isolate in pupae) to AMP. Crude peptides, including AMPs, were extracted from 2nd instar larvae, 4th instar larvae, and pupae, respectively. Neither the growth of *E. xiangfangensis* nor that of *E. innesii* was affected by AMPs isolated from 2nd instar larvae. Interestingly, growth inhibition of *E. innesii* was clearly seen when it was incubated with 4th instar larval and pupal AMPs (Fig. 6C). The *E. innesii* strain was isolated from larvae, and AMP antibacterial activity against this organism occurred with AMPs extracted only from pupae, thus confirming that AMP production, associated with particular metamorphic changes, induces changes in the intestinal microbial community.

**Changes in AMP-related gene expression following *Enterococcus* inoculation.** The data above confirmed that the pupation of *G. mellonella* changed according to the presence or absence of *Enterococcus* and that inhibition of *Enterococcus* growth occurred only in the presence of AMPs isolated from pupae. To identify *E. innesii*-specific AMPs, therefore, the expression of AMP-related genes was assayed in *G. mellonella* treated with vancomycin and then inoculated with *E. innesii*. This confirmed that the expression of genes coding for gallerimycin, cecropin, and IMPI (insect metalloprotease inhibitor) rapidly increased but was undetectable after 24 h of treatment with vancomycin. In contrast, following inoculation with *E. innesii*, the increased expression of these genes was maintained even 48 h after inoculation. On the other hand, apolipophorin gene expression was not observed following treatment with vancomycin, but increased following inoculation with *E. innesii* (Fig. 7).

## DISCUSSION

Changes in the gut microbiome profile according to developmental stage have been documented in a variety of Animalia kingdom organisms (6–10). Distinct changes in the gut microbiota during *G. mellonella* metamorphosis raise the possibility that the host microbial community may directly play a role in the developmental process. Changes in bacterial gut flora composition between larvae and pupae, in particular, demonstrate a dramatic transition between diderms (Gram negative) and monoderms (Gram positive).

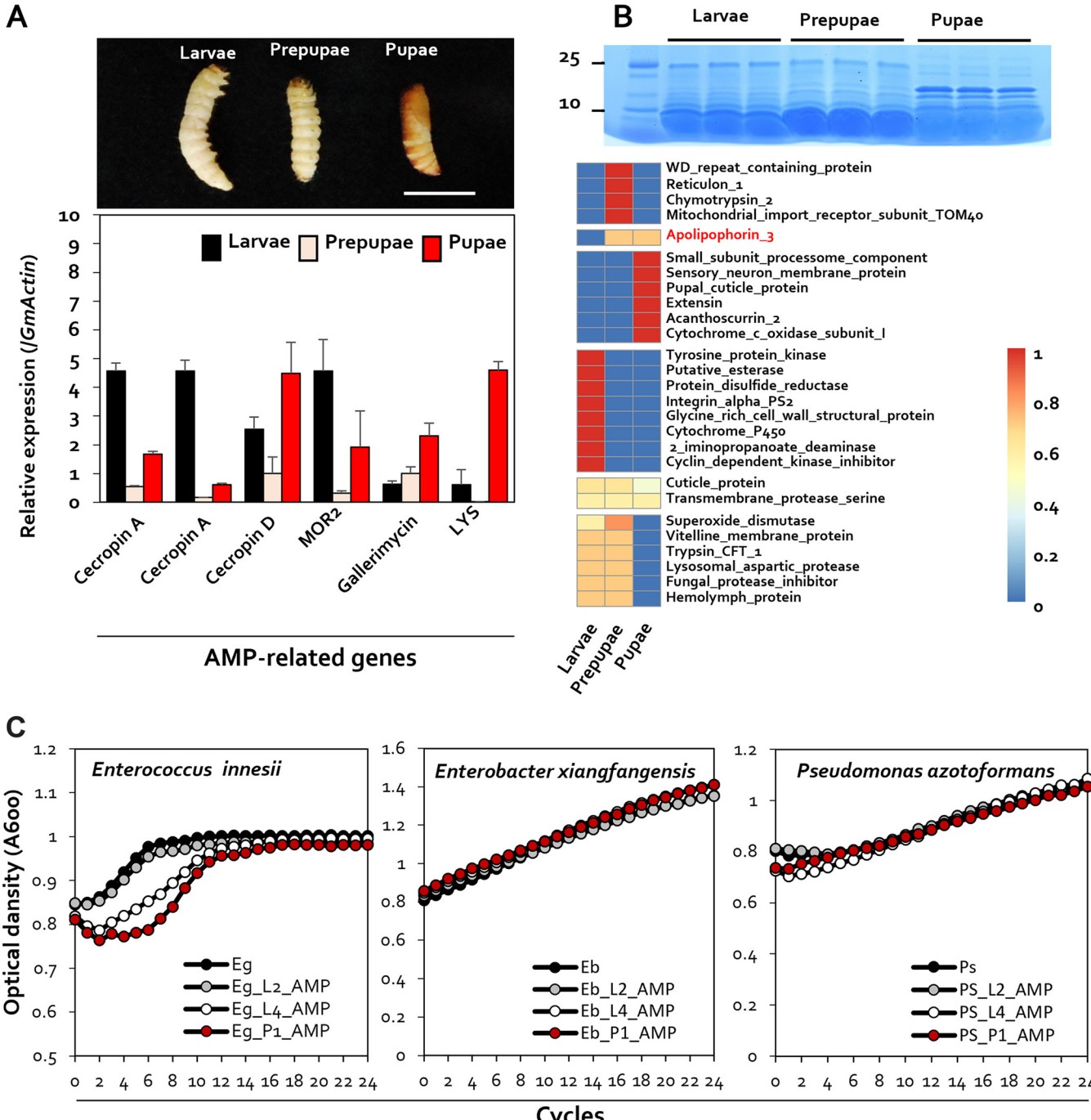

**FIG 6** Expression of *G. mellonella* antimicrobial peptides at different developmental stages. (A) Quantification of gene expression of the antimicrobial peptide genes coding for cecropin A, cecropin D, MOR2, gallerimycin, and LYS in larvae, prepupae, and pupae. The housekeeping gene coding for *G. mellonella* actin (*GmActin*) was used for normalization. Error bars represent the mean ± standard error of the mean (SEM). Sample size: $n = 5$ *G. mellonella* larvae per treatment. (B) SDS-PAGE analysis of peptide patterns for proteins of 25 kDa or less in the larval, prepupal, and pupal stages of *G. mellonella*. Shown is a heat map of differentially expressed proteins from larvae, prepupae, and pupae. Proteomic analysis was performed via LC-ESI-Q-TOF, and final qualitative analysis was performed via MS/MS analysis and UniProt data. (C) Antimicrobial activity of 2nd instar larval, 4th instar larval, and pupal antimicrobial peptide extracts against *Enterococcus innesii*, *Enterobacter xianfgangensis*, and *Pseudomonas azotoformans*. The figure shows bacterial growth curves in the presence of antimicrobial peptides extracted from *G. mellonella* at different developmental stages.

Additionally, the lack of diversity in the bacterial composition of larval and pupal stages, compared with the egg stage, shows that the presence of certain bacterial species may play a functional role in the metamorphosis process (Fig. 1). A simple cultivation test demonstrated that the gut flora of larvae and pupae is dominated by *E. innesii* (a monoderm)

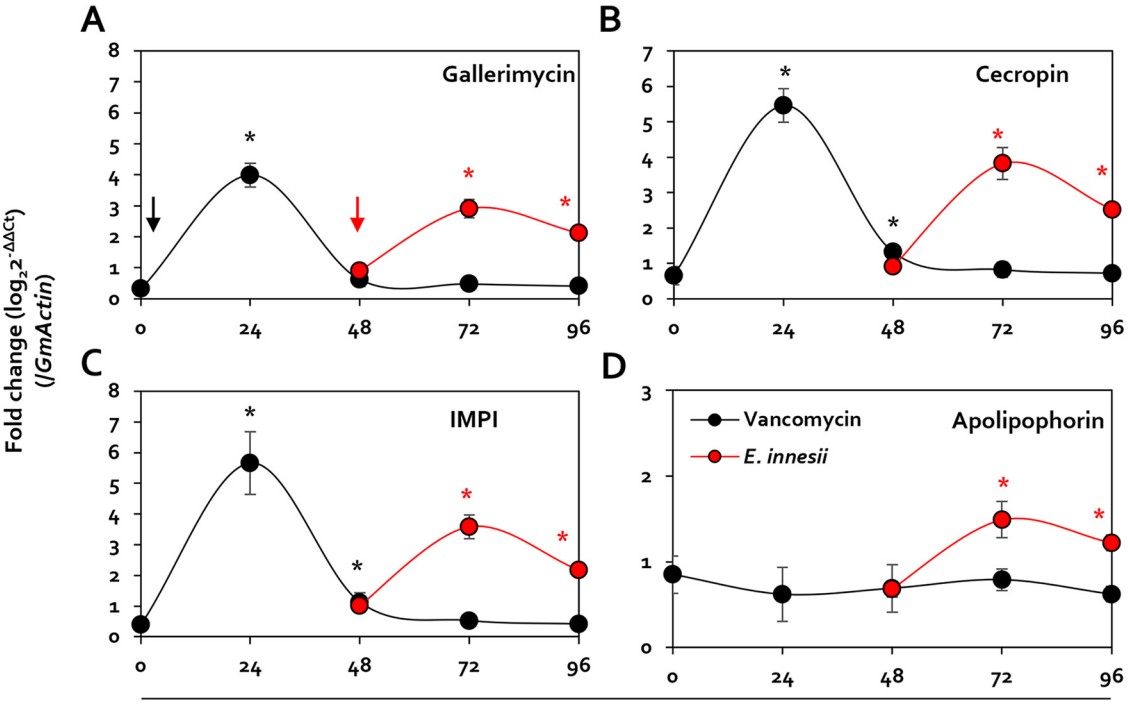

**FIG 7** Analysis of gene expression changes in larvae, following vancomycin treatment, through real-time quantitative PCR (RT-qPCR). Shown are levels of expression of the IMPI, cecropin D, gallerimycin, and apolipophorin III genes, which increase in larvae after being fed with vancomycin, shown as fold change (log$_2$ threshold cycle [$2^{-\Delta\Delta CT}$]) by RT-qPCR. The gene coding for $\beta$-actin was used as the reference gene. *E. innesii* was treated 48 h after antibiotic treatment, and antibiotic treatment and *E. innesii* treatment are indicated by black and red arrows, respectively.

and *P. azotoformans* (a diderm), respectively (Fig. 2). The discrepancy between the dominant genera identified in pupae by 16S rRNA amplicon sequencing (*Enterobacter*) and by culture (*Pseudomonas*) could be explained in one of two ways: (i) strains of the actual dominant genus (*Enterobacter*) in pupae were not recovered using the cultivation technique we employed in this study, or (ii) diderms are irrelevant to the developmental process, and so the predominance of diderms might simply be the consequence of a marked reduction in *Enterococcus*. Larvae challenged with polymyxin B showed no change in their pupation rate, demonstrating that diderms do not play a significant function in the developmental process of *G. mellonella* (Fig. 3B). On the other hand, vancomycin challenge of larvae resulted in a notable outcome in which the pupation rate was slowed in the absence of *E. innesii* and, presumably, all other monoderms. When antibiotic-treated larvae were reinoculated with *E. innesii* alone, the pupation rate was the same as that with the mock treatment. These findings show that intestinal *E. innesii* is required for metamorphic transformation (Fig. 3).

In addition, the restoration of pupation solely in the presence of *Enterococcus*, but not other Gram-positive bacteria, provides evidence for the specificity of the response to *Enterococcus* (Fig. 3). Furthermore, when confirming what factors caused the specific response to *Enterococcus*, it was found that the recovery of pupation occurred in the presence of cell debris of *Enterococcus* (Fig. 4). These results suggest that control of the concentration of bacteria in the intestine of *G. mellonella* may be mediated through detection of membrane components of *Enterococcus*.

Hormones and the immune system of animals are critical not only in preventing harmful bacterial invasion but also in modulating the composition of the gut microbiota (25, 26). Transcriptome analysis revealed that genes involved in hormone production were upregulated in the larval stage, whereas genes involved in the immune response were upregulated in the pupal stage (Fig. 5). The demonstration that there is

significant production of AMP-related peptides only in the pupal stage suggests that alterations in intestine microbial populations may occur as a result of metamorphosis-induced host genetic changes. Recent research on intestinal AMPs demonstrated their critical role in controlling the gut microbial ecology and promoting positive homeostasis (27). Cecropin A and cecropin D are both members of the cecropin family, and both AMPs exhibit potent activity against Gram-positive and Gram-negative bacteria (28, 29). Gallerimycin has also been found to be associated with the formation of larval intestinal bacteria. Distinct profiles of AMPs are produced during different developmental stages, and those produced in 4th instars and pupae can affect the reduction of *Enterococcus innesii in vitro* (Fig. 6) In particular, when the expression of pupa-specific genes in *G. mellonella* was measured after treatment of larvae with vancomycin, the expression of genes encoding IMPI, cecropin D, and gallerimycin, which are known to have increased expression in pupae, increased significantly (Fig. 7; see Table S1 in the supplemental material). These changes in AMP expression in larvae, following antibiotic treatment, suggest that the presence of Gram-positive bacteria in the intestine correlate specific patterns of gene expression in pupae.

In addition, after treatment with vancomycin, pupation rates were restored only with species of *Enterococcus* routinely found in the intestine of *G. mellonella*. Furthermore, inoculation of pupae with *E. innesii*, following antibiotic treatment, resulted in rapid increases in the expression of genes related to expression of gallerimycin, cecropin, and IMPI, but their expression decreased to undetectable levels after further vancomycin treatment (Fig. 7). This suggests the possibility that the three genes could be induced in response to transient stimuli or stress. However, it can be confirmed that the increase in gene expression after feeding additional *E. innesii* does not disappear temporarily, unlike vancomycin, and is maintained up to 96 h (Fig. 7). On the other hand, in the case of apolipophorin, vancomycin treatment did not show gene expression, but it was confirmed that the expression was specifically induced in *E. innesii*.

These data suggest that specific interaction between the Gram-positive bacteria present in the intestine and host metamorphosis regulation has occurred. The mechanisms by which *G. mellonella* controls its intestinal microbial flora was explored at the genetic level and with reference to the importance of the developmental stage of the insect in this control. Transitions in the intestinal microbial community associated with metamorphosis are an intriguing topic in host-microbial interaction studies. However, the majority of studies, to date, have focused on intestinal microbe cluster analysis, which alone does not provide light on the function of the host and/or intestinal microorganisms in the metamorphosis processes. To address these links at each of the developmental stages of metamorphosis, we used NGS to analyze *G. mellonella* gene expression alongside microbiome analyses. We found that different degrees of pupation occurred, depending on the presence or absence of *E. innesii* strains. Additionally, we hypothesize that *G. mellonella* utilizes AMPs to exert control over gut microorganisms.

Our study confirmed the important role of the gut microbiota in host ecological changes during the host life cycle (Fig. 8). Oral injection of *Enterococcus innesii* isolated from larval intestines induced significant changes in insect metamorphosis. The significance of this study is that it not only confirmed the inhibition of metamorphosis by *E. innesii*, which is recognized as an insect symbiotic bacterium, but it also revealed differences in the patterns of host-produced AMPs (and thus their bacteriostatic effect) according to the host stage. In general, the influence of the host on the composition of its gut microbiota has focused on host immunological factors. Our new findings suggest that there are other host-specific mechanisms that influence the gut microflora and, in turn, metamorphic developmental changes.

## MATERIALS AND METHODS

***Galleria mellonella* rearing and treatment.** *Galleria mellonella* larvae were purchased from Ecowin (Andong, South Korea). Each developmental *G. mellonella* larva was collected, and the surface was rinsed with 100% ethanol and sterile water. Each larva was reared in the dark in a 6.5-L plastic cage (31.5 by 23 by 11.5 cm) (Lock & Lock Co., Ltd., South Korea) filled with nutrient-rich food at 30°C. The food consisted

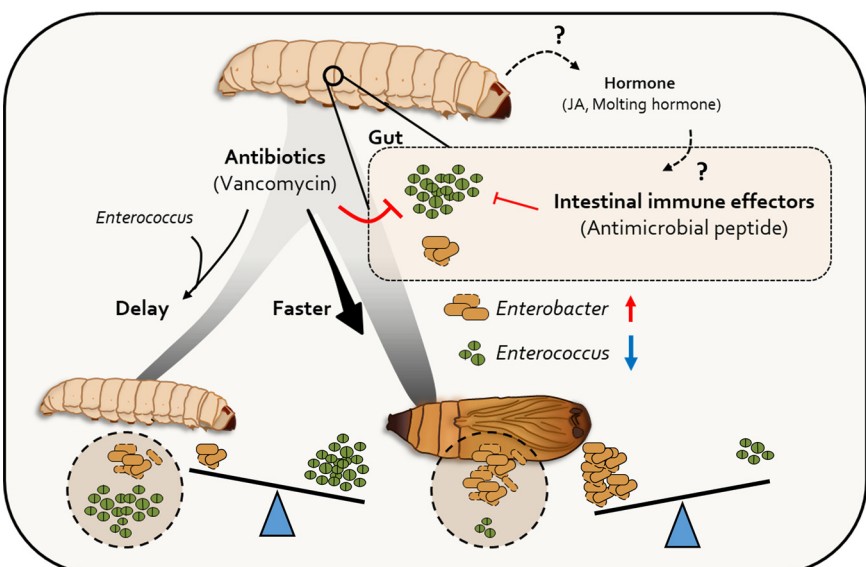

**FIG 8** Role of gut microbiota during metamorphosis. The transition of gut microbiota accelerates the pupation rate. Antimicrobial peptide production, which influences gut microbiota transition, is programmed as a biological clock at the early pupation stage. Antimicrobial peptides reduce the level of *Enterococcus*, resulting in the induction of *G. mellonella* pupation.

of 300 g wheat bran, 300 g rice bran, 2.25 g yeast extract, 1 g $CaCO_3$, 125 mL glycerol, 87.5 mL honey, and 300 mg vitamin B complex. Food was replaced once a week.

All larval gut samples were taken in triplicate. To remove intestinal enterococci, 10 $\mu$L of 100 $\mu$g/mL vancomycin (Sigma-Aldrich) was administered to the 4th instar larvae by a force-feeding method. All larvae were normalized at 15°C for 12 h prior to administration of antibiotics, and the presence or absence of microbial removal was confirmed by dissecting the intestines of randomly selected larvae and incubation on tryptic soy agar (TSA) at 30°C for 2 days. To correlate metamorphic changes in larvae with specific bacterial strains, intestinal microorganisms were first removed by antibiotic treatment and then retreated with antibiotics or reinfected with the isolated bacterial strains 3 days later. All experiments were performed in triplicate on 30 insects.

In order to confirm the elimination of *Enterococcus* from the intestine by vancomycin treatment, a rifampin resistance (Rif$^r$) marker was introduced into *E. innesii* and *E. mundtii* isolates from the intestine of *G. mellonella*. Each Rif$^r$-marked bacterium was adjusted to a concentration of optical density at 600 nm (OD$_{600}$) of 1.0 in PBS, and 10 $\mu$L was injected into *G. mellonella*, followed by incubation at 30°C for 24 h. Vancomycin (10 $\mu$L of a 100-$\mu$g/mL solution) was orally administered, and after 24 h, the intestines were removed and used for enumeration of the two bacteria by plating on TSA containing rifampin (100 $\mu$g/mL). *G. mellonella* administered with PBS was used as a negative control.

**Preparation of bacterial inocula.** Bacterial strains were cultured in tryptic soy broth (TSB) at 30°C for 12 h with shaking at 150 rpm. Cells were recovered from the culture, washed in PBS, and administered to larvae via the force-feeding method at a final concentration of $1 \times 10^5$ CFU/larva. PBS treatment was used as a control. All treatments measured the degree of conversion to pupae per day at 30°C. The TSB culture medium was centrifuged at 13,000 rpm for 5 min and separated into extracellular components (ECs) and cell pellets. In addition, after suspension of the cell pellet in sterilized water, bacterial cell lysis was performed using ultrasonication. The ultrasonication conditions were 10 s on and 10 s off for a total of 1 min with the pulser power set at 20%. After centrifugation at 13,000 rpm for 5 min, intracellular components (ICs) and cell debris were obtained and used for inoculation experiments.

**16S rRNA amplicon pyrosequencing.** Eggs, larvae, pupae and adults were sampled and washed twice with 100% ethanol. Triplicate samples were taken for each insect stage. At each stage, *G. mellonella* populations were collected and rinsed with 100% ethanol and sterile water. For eggs, DNA isolation was performed after surface sterilization with 100% ethanol, and for larvae and pupae, gut samples were collected for subsequent analysis. DNA isolation from each sample was performed using a genomic DNA isolation kit (Wizard, USA). PCR amplification was carried out using primers targeting the V1 to V3 regions of the 16S rRNA gene. Sequencing was performed at Theragen (Bio Institute, Suwon, South Korea) and processed by the GS Junior sequencing system (Roche, Branford, CT, USA) according to the manufacturer's instructions. The readings obtained from each sample were sorted by the unique barcode of each sequence product. All readings containing two or more ambiguous nucleotides, low-quality scores (average score of <25), or readings below 300 bp were discarded. Potential chimeric sequences were removed by the Bellerophone method and then taxonomically assigned via the EzTaxon-e database (http://eztaxon-e.ezbiocloud.net). The richness and evenness of the samples were determined by ACE, Chao1, Jackknife, NPShannon, Shannon, and Simpson indices at 3% distance.

**LEfSe analysis.** LEfSe analysis was performed using Huttenhower Galaxy server (http://huttenhower.sph .harvard.edu/galaxy/), and the analysis was conducted with multiclass analysis with all against all. The $\alpha$ value

was divided by a 0.05 Kruskal-Wallis test between classes, and a 0.05 pairwise Wilcoxon test was performed. In addition, the logarithmic LDA score was cut off at 2.0 for discrimination of each microorganism.

**Isolation and identification of intestinal microbes.** The intestines of *G. mellonella* larvae were harvested for the isolation and identification of culturable bacteria. For pupae, their surfaces were sterilized twice in 100% (vol/vol) ethanol, and the whole body was ground through bead beating for 10 s. Each sample was suspended in PBS and cultured on TSA to generate individual bacterial colonies. Ten bacterial colonies were randomly selected and identified. The whole-genome sequence of *Enterococcus* used in the experiment was reported by Chung et al. (22). Whole-genome sequences of the type strains of the genus *Enterococcus* and *Enterococcus* sp. strain CR-Ec1 were obtained from NCBI database (https://www.ncbi.nlm.nih.gov/). The average nucleotide identity (ANI) value between *Enterococcus* CR-Ec1 and other type strains was calculated using JSpeciesWS (https://jspecies.ribohost.com/jspeciesws/), and ANI-based phylogeny was constructed using the Orthologous Average Nucleotide Identity Tool (CJ Bioscience, South Korea) with species with high ANI values. The isolated bacteria were stored at −80℃ and used in subsequent experiments. All larval and pupal gut samples were taken in triplicate.

**_G. mellonella_ RNA isolation.** Total RNA was isolated from larvae and pupae using TRIzol reagent (Invitrogen). The combination of TRIzol reagent and each sample was ground using bead beating in a 1.5-mL centrifuge tube. RNA was precipitated with isopropanol and washed with 75% (vol/vol) ethanol according to the manufacturer's instructions. Total RNA was suspended in 50 $\mu$L of 0.1% (vol/vol) diethyl pyrocarbonate (DEPC) and stored at −80℃.

**Transcriptomic analysis of _G. mellonella_.** Using the TruSeq stranded total RNA kit, transcriptome sequencing (RNA-seq) data were generated from the larvae and pupae of *G. mellonella*. Bowtie2 was used to map 21 RNA-seq samples to *G. mellonella* assemblies. After indexing, the raw data were mapped to the assembly using TopHat2 as the default setting. Expression values in the mapped BAM file were calculated using Cufflinks. To reduce the bias between samples, the expression values were normalized using a standardization method. In addition, for accurate identification of differentially expressed genes (DEGs), during larval development, a *t* test was performed using Welch's *t* test method and DEGs were selected when the FDR was <0.05.

**Sparse canonical correlation analysis.** Sparse canonical correlation analysis (CCA) was performed to analyze the correlation between microbiome data and RNA-seq expression in the larval and pupal stages of *Galleria mellonella*. The microbiome database used the results obtained from the qiime2 analysis, and after removing a taxon with a read number of 0 in the two treatment groups, a taxon with a difference of more than two times was used. In the case of RNA-seq data, the DEG analysis results were used, genes with 0 in both treatment groups were removed, and a gene list with a difference of more than 2-fold between treatment groups was performed. CCA was performed using the iSFun package of the R program. In the obtained data, the gene list was grouped by performing Gene Ontology analysis using PANTHER GO-Slim. The results were displayed as a heat map using the R program.

**Purification and characterization of AMPs.** Two hundred milligrams of larvae and pupae was immersed in 1 mL of methanol-glacial acetic acid-water (90:1:9 [vol/vol/vol]) and thoroughly ground using bead beating. The precipitated protein was pelleted by centrifugation at 20,000 $\times$ *g* for 30 min at 4℃. The resulting supernatant was collected, and the peptide was dried using a vacuum concentrator (SA-VC-300H) and stored at −20℃ until needed.

To investigate the antimicrobial growth inhibition effect of crude peptide extracts, including AMPs, on intestinal bacteria from *G. mellonella*, bacteria were adjusted to an $OD_{600}$ of 0.1 and treated with the AMP at a 1/100 dilution, respectively. The $OD_{600}$ measurements for bacterial cultures were performed in a Tecan M200 fluorescent spectrophotometer, using a Costar flat-bottom 24-well plate with a lid and 1 mL per well for all measurements.

**Proteomic analysis of _G. mellonella_ during metamorphosis.** Proteins were extracted from the larval, prepupal, and pupal stages of *G. mellonella* and the patterns of the peptides were confirmed by 12% SDS-PAGE. For profiling of AMPs, an in-gel piece of 25 kDa or less was decolorized by reacting in 50% $CH_3CN$ ($H_2O$) solvent for 15 min. Trypsin solution (sequencing-grade modified porcine trypsin; 1 $\mu$g in 50 $\mu$L) (Promega, Madison, WI, USA) was added to decolorized in-gel piece samples and incubated at 37℃ for 16 h. Dithiothreitol (DTT) (2 $\mu$L of 1 M DTT) (GE) was added, and the mixture was incubated at room temperature for 1 h. Iodoacetamide (IAA) (4 $\mu$L of 1 M IAA) (Sigma) was added and reacted at room temperature for 1 h. The gel was dehydrated by adding 50% $CH_3CN$ ($H_2O$) solvent (200 $\mu$L), and the hydrolyzed peptide was recovered and dried using a SpeedVac.

**LC-ESI-Q-TOF-MS analysis.** The hydrolyzed peptides were subjected to qualitative analysis via LC-ESI-Q-TOF-MS (ABsciexTripleTOF 5600+). The chromatograph used was a Thermo (Dinex) UHPLC Ultimate 3000, and the columns were Acquity ultraperformance liquid chromatography (UPLC) BEH 130-Å $C_{18}$ columns. The Acetonitile: 1% formic acid solvent used was 99: 1 for 3 min, 50:50 for 3 to 70 min, 0: 100 for 80 min, 0: 100 for 80 min at a rate of 300 $\mu$L/min. The fractionated peptides were subjected to final qualitative analysis via tandem MS (MS/MS) analysis and UniProt data (www.uniprot.org).

**Statistical analysis.** The results were obtained by performing one-way repeated-measures analysis of variance (ANOVA) using the Bonferroni posttest and two-tailed Student's *t* test using the R program.

**Data availability.** Sequencing data have been deposited in the NCBI Sequence Read Archive (SRA) database under accession no. PRJNA386430.

## SUPPLEMENTAL MATERIAL

Supplemental material is available online only.

**SUPPLEMENTAL FILE 1**, PDF file, 1.4 MB.

## ACKNOWLEDGMENTS

This work was supported by funds derived from the National Research Foundation of Korea (NRF) funded by the Korean government (MSIT) (NRF-2022R1C1C1002780) for J.-S.K. and the National Research Foundation of Korea (NRF) funded by the Ministry of Science and ICT (NRF-2020M3E9A1111636 and NRF-2021M3A9I5021439), Center for Agricultural Microorganism and Enzyme (Project No. PJ015049) of Rural Development Administration, and the KRIBB Initiative Program, South Korea, for C.-M.R.

J.-S.K. and C.-M.R. designed the study. H.G.K., J.-H.C., S.L., and J.-S.K. performed the experiments. H.G.K., J.-S.K., and C.-M.R. analyzed the data. J.-S.K., C.-M.R., and H.G.K. wrote the paper.

We declare no conflict of interest.

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
