## [Reviewer comments · Microbiology Spectrum]

Microbiology Spectrum

Population dynamics of intestinal *Enterococcus* modulate *Galleria mellonella* metamorphosis

Hyun Gi Kong, Joon-Hwui CHUNG, Soohyun Lee, Jun-Seob Kim, Choong-Min Ryu, and Jin-Soo Son

Corresponding Author(s): Choong-Min Ryu, Korea Research Institute of Bioscience and Biotechnology

Review Timeline:

Submission Date:	July 20, 2022
Editorial Decision:	October 25, 2022
Revision Received:	February 28, 2023
Editorial Decision:	April 10, 2023
Revision Received:	May 23, 2023
Accepted:	May 24, 2023

Editor: Vittal Ponraj

Reviewer(s): Disclosure of reviewer identity is with reference to reviewer comments included in decision letter(s). The following individuals involved in review of your submission have agreed to reveal their identity: Daniel F. Q. Smith (Reviewer #2)

Transaction Report:

DOI: <https://doi.org/10.1128/spectrum.02780-22>

October 25, 2022

Dr. Choong-Min Ryu
Korea Research Institute of Bioscience and Biotechnology
Infectious Disease Research Center
125, Gwahak-ro, Yuseong-gu
Daejeon 34141
Korea (South), Republic of

Re: Spectrum02780-22 (Population dynamics of intestinal Enterococcus modulates Galleria mellonella metamorphosis)

Dear Dr. Choong-Min Ryu:

Link Not Available

Sincerely,

Vittal Prakash Ponraj Ph.D., SM(ASCP)CM

Journals Department
Reviewer comments:

Reviewer #1 (Comments for the Author):

1. Line 22: Is there a reason pyrosequencing was used versus more current methods?
2. Abstract says V4 region, but methods say V1-V3. These will produce different results for microbiome data when binning, please clarify
3. The methodology is unclear on how the strains pulled out of the host were speciated. Where they just sequenced and blasted? 16S region only, or full sequence? Your phylogenetic tree is not convincing for speciation. Did you do additional testing? MALDI-TOF? Biochemical assays?
4. Line 108-109: prevalent would not be the correct wording given you already stated 50% of the reads mapped to Enterobacter genus, and this implies absoluteness not a comparison/or relativeness. In order to properly interpret these results and word this, one would have to know in the methods is you used all-against-all or the one-against-all feature detection mode of LefSe when

- comparing multiple groups. For the all-against-all, a feature has to be differentially abundant between all the 3 groups. Eg. Feature X is detected if it is differentially abundant in A compared to B, in A compared C, AND in B compared to C. In contrast for one against all: Feature X is detected if it is differentially abundant in A compared to B, and A compared to C BUT NOT necessarily in B compared to C. The terminology usually used is "enriched" in whatever group you are referring to...
5. Line 114-117: Are these absolute reads or rarified?
 6. Line 169-173. This is overstated. This is association, not causation. There is no confirmation here by just looking at upregulation/downregulation of genes between in pupae and larvae stages that suggests a connected mechanism for tissue reorganization in pupation, AMPs, and the intestinal microflora.
 7. Lines 181-185. Since these are expression of genes, you should list the gene name and what it potentially encodes when discussion expression analyses, not the protein product it makes. Transcription does not necessarily mean translation, or if there are protein modifications.
 8. Lines 198-209. These experiment described here does not investigate whether AMP produced during the pupal stage of the development of *G.mellonella* influences the composition of the intestinal microbiota or induces changes in the intestinal microbial community, as stated. What this experiment does is examine the effect of AMP on the growth of *E. gallinarum*, *E. xiangfangensis*, and *P. azotoformans*. Although they suggest they are the main constituents, it would be better to see what those AMPs do to the total microbial community in those stages. As even small changes to lesser represented constituents of the microbiome can have impactful effects. If the total microbial community is not tested, then reword this paragraph to accurately reflect the experiment performed.
 9. Lines 222-227: Another explanation is that you based speciation on what seems to be only sequence homology. Basing strain identification on sequence homology is tricky as you are reliant on good databases and assuming your strains are represented. Some of these organisms have high rates of recombination (particularly *Enterococcus*) meaning they can map to closely related species. Further methods for identification should be used before making these this assumptions.
 10. Lines 227-229... All results even if supplemental should be discussed in the results, as we have no context for how and why this experiment was done.
 11. Line 251-255: Some network analyses of other statistical methodology could be presented to see the correlations between the *G. mellonella* gene expression with the microbiome at each stage. Also since this is a linear vent, one could do linear mixed model or time dependent models to look at the change in gene expression, and change in microbiome overtime to see how they are correlated.
 12. Line 328...with the respective/target AMP?? You tested more than one, yes?
 13. Main paper says LC/MS..methods says LC-ESI-QTOF-MS/MS, these are two different methods, please clarify
 14. Table 1: Have other papers in *G. mellonella* reported such low #s for OTUs in larval stage??
 15. Table 1: Probably best to define 3/4-YBK in table footnote
 16. Figure 2: Please label graphs appropriately in panel A. The 16S reads only map to the genera level, not the species level
 17. Figure 2 B, to limit confusion, I would color *Enterobacter* and *Pseudomonas* differently. They are very different bacteria and should not be lumped together.
 18. What statistical test is used in Figure 3?? State in methods or figure legend

Reviewer #2 (Comments for the Author):

This work by Kong et. al., seeks to evaluate whether the midgut microbiome of the *G. mellonella* wax moth controls its metamorphosis. They show that the bacteria species *Enterococcus gallinarum* is uniquely associated with larval stages of *G. mellonella*. When the bacteria are depleted using antibiotics there is a rapid onset of pupation, and this rapid pupation is rescued by the reintroduction of *E. gallinarum*. The authors claim that the onset of pupation via hormone production induces the expression of antimicrobial peptides in response to the *E. gallinarum* present in the gut, which controls the level of *E. gallinarum* in the midgut, which then results in onset of pupation due to bacterial depletion.

The story presented by the authors is very interesting and could provide important insight into how bacteria regulate or are intimately involved in the *G. mellonella* lifecycle. Further, the model presented is insightful, and illustrated beautifully in Figure 6. However, I have some hesitations regarding the conclusions drawn from the data presented, mainly due to it seemingly being more correlative than causative, while claiming causation.

Figures 1 through 3 are important for establishing the midgut microbiome that is present during different life stages and showing that depletion and re-addition of specific members of the microbiome affects the time until pupation. The authors convincingly show that treatment with antibiotics, and then re-introducing bacteria into the midgut speeds up and restores the rate of pupation, respectively. Figures 4 and 5 show what genes and antimicrobial peptides are expressed during different life stages, which correlates to changes in the midgut microbiome. While they show that purified AMPs from the larval stages selectively kill the *E. gallinarum*, they do not show a causative link or directionality between microbiome composition and expression of antimicrobial peptides, as they suggested through a feedback loop mechanism. This can be accomplished by looking at gene expression or AMP expression following antibiotic treatment and/or re-introduction of the microbiome. They do not show that the AMP/hormone expression is changed when the *E. gallinarum* is absent, which seems to be a major feature of their feedback loop mechanism.

I think the conclusion that *E. gallinarum* exclusively affects pupation rate is not completely proven. While re-addition of *E. gallinarum* alone in the midgut of the larvae convincingly restores normal pupation rate following antibiotic treatment, it is not clear whether the effect is due specifically to the presence of *E. gallinarum* or due to the restored levels of any *Enterococcus* spp./bacteria present in the midgut. It is unclear whether re-introduction of large quantities of other *Enterococcus* species or an unrelated gram-positive bacterium would have similar effects.

Additionally, the authors did not confirm whether the midgut microbiome was depleted by the addition of vancomycin. While it would make intuitive sense, it would be good to confirm, in case the drug is not as effective or there are persistent drug-resistant *Enterococcus* spp still present.

I found the Materials and methods section to be lacking information on some of the experiments performed, namely:

- Rearing conditions of the larvae that were purchased, and what they were fed once they hatched from the eggs.
- Definition of TSA and TSB media.
- How egg and adult microbiomes were collected.
- Definition of how instars were differentiated.
- Were the AMP activity experiments done on whole *Galleria* protein extract? The section beginning at Line 322 does not describe purification of the AMP (although the section beginning on line 332 does somewhat describe that purification process). Was the same purification of small peptides via a gel performed, or were bacteria just treated with all the protein from homogenized *G. mellonella*?

Specific comments:

Abstract: I found this section to be a little too dry and written in passive voice.

Line 37: the directionality discussed in this sentence is not well explained

Introduction: It might be good to start off with an insect microbiome introduction paragraph.

Line 68: Change "adipose" to "fat"

Do you think the microbial composition of the eggs is related to the microbes from the environment (container, food) or from the reproductive tract of the female moth?

Figure 1: Define the instar abbreviations.

Line 101 to 104: Not sure this is a logical conclusion from the data in Figure 1, seems more like the conclusion from the subsequent figures.

Figure 4: It is unclear what the expression levels are being compared to. Fold change compared to what conditions/life stage? Pupae to larvae, larvae to pupae? What about when both are shown? There might be a better way to display or explain what is being shown.

Lines 245-246: I do not think these claims are supported, as there is only *in vitro* evidence that the AMPs kill *E. gallinarum*, not that they actually control *E. gallinarum* levels in the larval midgut.

Discussion: All data should be originally introduced in the results section, including the supplementary figures. Figure S3 is nice data and could be incorporated into one of the main figures. n

Figure 6 is very impressive and well-illustrated!

Staff Comments:

Preparing Revision Guidelines

- Point-by-point responses to the issues raised by the reviewers in a file named "Response to Reviewers," NOT IN YOUR COVER LETTER.

- Upload a compare copy of the manuscript (without figures) as a "Marked-Up Manuscript" file.
- Each figure must be uploaded as a separate file, and any multipanel figures must be assembled into one file.
- Manuscript: A .DOC version of the revised manuscript
- Figures: Editable, high-resolution, individual figure files are required at revision, TIFF or EPS files are preferred

Please return the manuscript within 60 days; if you cannot complete the modification within this time period, please contact me. If you do not wish to modify the manuscript and prefer to submit it to another journal, please notify me of your decision immediately so that the manuscript may be formally withdrawn from consideration by Microbiology Spectrum.

Point by Point Response

[Reviewer 1]

1. Line 22: Is there a reason pyrosequencing was used versus more current methods?

Response: Our data were analyzed using pyrosequencing technology before the development of Miseq technology. However, the microbial diversity of Galleria's intestinal environment used in the study is very simplistic. Therefore, we do not think that the use of pyrosequencing will make a difference to our results. In addition, being able to sequence long reads with pyrosequencing rather than elevating deeps with Illumina Miseq could be an advantage in this study.

2. Abstract says V4 region, but methods say V1-V3. These will produce different results for microbiome data when binning, please clarify

Response: Sorry. It was mistake. The regions we used were changed in the text as V1-V3. P2 L22

3. The methodology is unclear on how the strains pulled out of the host were speciated. Where they just sequenced and blasted? 16S region only, or full sequence? Your phylogenetic tree is not convincing for speciation. Did you do additional testing? MALDI-TOF? Biochemical assays?

Response: Thanks for the advice. Further species classification was performed with whole-genome data of Enterococcus isolated from *Galleria mellonella* larvae, and as a result, *Enterococcus innesii* was finally identified. Relevant results have been corrected in Fig2C.

The related results were modified as follows in the text.

“Species classification was performed for Gram-positive bacteria, which are dominant in *G. mellonella* larvae. Whole genome analysis of bacteria isolated from larvae was performed through previous studies (ref), and when identified based on relevant whole genome data, it was finally identified as *Enterococcus innesii* (Fig. 2C).” P8 L133-137

In addition, the related reference papers below have been added.

“Chung J-H, Jeong H, Ryu C-M.2018. Complete Genome Sequences of Enterobacter cancerogenus CR-Eb1 and Enterococcus sp. Strain CR-Ec1, Isolated from the Larval Gut of the Greater Wax Moth, Galleria mellonella. Genome Announc. 15:e00044-18.”

4. Line 108-109: prevalent would not be the correct wording given you already stated 50% of the reads mapped to Enterobacter genus, and this implies absoluteness not a comparison/or relativeness. In order to properly interpret these results and word this, one would have to know in the methods is you used all-against-all or the one-against-all feature detection mode of LefSe when comparing multiple groups. For the all-against-all , a feature has to be differentially abundant between all the 3 groups. Eg. Feature X is detected if it is differentially abundant in A compared to B, in A compared C, AND in B compared to C. In contrast for one against all: Feature X is detected if it is differentially abundant in A compared to B, and A compared to C BUT NOT necessarily in B compared to C.

Response: Thanks for the advice. We performed LefSe using the Huttenhower Galaxy Server (<http://huttenhower.sph.harvard.edu/galaxy/>), and the analysis was conducted with multi-class analysis all-against-all. Alpha value was divided by 0.05 ruskal-Wallis test between classes, and 0.05 pairwise Wilconxon test was performed. In addition, the logarithmic LDA score was cutoff at 2.0 for discrimination of each microorganism. Added related information to method.

“LEfSe analysis was performed using Huttenhower Galaxy Server (<http://huttenhower.sph.harvard.edu/galaxy/>), and the analysis was conducted with multi-class analysis all-against-all. Alpha value was divided by 0.05 ruskal-Wallis test between classes, and 0.05 pairwise Wilconxon test was performed. In addition, the logarithmic LDA score was cutoff at 2.0 for discrimination of each microorganism.” P20 L390-394

The terminology usually used is "enriched" in whatever group you are referring to...

Response: Thanks for the detailed advice. We agree with what you said, and have modified “prevalent” to “enriched” in the manuscript. P7 L111

5. Line 114-117: Are these absolute reads or rarified ?

Response: Yes, that are the absolute read numbers.

6. Line 169-173. This is overstated. This is association, not causation. There is no confirmation here by just looking at upregulation/downregulation of genes between in pupae and larvae stages that suggests a connected mechanism for tissue reorganization in puparation, AMPs, and the intestinal microflora.

Response: Thank you. I also agree with what you said. Therefore, the contents of the main text have been modified.

“Therefore, our result suggests that there can be a correlation between these immune response changes and the expression of AMPs and changing the intestinal microflora.” P11 L210-212

7. Lines 181-185. Since these are expression of genes, you should list the gene name and what it potentially encodes when discussion expression analyses, not the protein product it makes. Transcription does not necessarily mean translation, or if there are protein modifications.

Response: Thanks for the advice. We modified the protein names in the text to gene names.

“HCM000600 and HCM009440 (Cecropin A) had an expression value of 4.58 ± 0.28 at the larval stage, higher than prepupal (0.53 ± 0.04 ,) and pupae (1.67 ± 0.09). On the other hand, HCM007113 (cecropin D), HCM008333 (gallerimycin), and HCM003020 (LYS) were 4.48 ± 1.09 , 2.30 ± 0.46 , and 4.59 ± 0.31 respectively in the pupal stage, which was higher than the expression values in the larval stage such as 2.54 ± 0.42 , 0.63 ± 0.11 , 0.61 ± 0.52 respectively (Fig. 6A).” P11 L221-225

8. Lines 198-209. These experiment described here does not investigate whether AMP produced during the pupal stage of the development of *G.mellonella* influences the composition of the intestinal microbiota or induces changes in the intestinal microbial community, as stated. What this experiment does is examine the effect of AMP on the growth of *E. gallinarum*, *E. xiangfangensis*, and *P. azotoformans*. Although they suggest they are the main constituents, it would be better to see what those AMPs do to the total microbial community in those stages.

Response: Thanks for the advice. In order to test the effect of AMPs on the entire microbial community, a system that can test only the effect of AMP is required, representing the microbial time at which AMPs is produced. Therefore, these aspects need to be clarified through follow-up studies.

As even small changes to lesser represented constituents of the microbiome can have impactful effects. If the total microbial community is not tested, then reword this paragraph to accurately reflect the experiment performed.

Response: Thanks for the advice. Agree on the possibility that a small number of microorganisms may have an effect. In our case, since it was confirmed that pupariation was restored when the *E. innesii* introduced orally into larvae, it was expected that changes in major bacteria at each stage would be important for pupariation, and the main bacteria were focused on AMP. Impact confirmed. Therefore, to reflect the experiment more accurately, the contents of the main text have been modified as advised.

“To investigate whether AMP produced during the pupal stage of the development of *G. mellonella* influence on the major intestinal bacteria at each stage, we examined the sensitivity of *E. innesii* (the major bacterial isolate in larvae), *E. xiangfangensis* and *P. azotoformans* (the major bacterial isolate in pupae) on AMP. The AMP was extracted from 2nd instar larvae, 4th instar larvae, and pupa, respectively.” P12 L238-242

9. Lines 222-227: Another explanation is that you based speciation on what seems to be only sequence homology. Basing strain identification on sequence homology is tricky as you are reliant on good databases and assuming your strains are represented. Some of these organisms

have high rates of recombination (particularly *Enterococcus*) meaning they can map to closely related species. Further methods for identification should be used before making these this assumptions.

Response: Response: Thanks for the advice. Further species classification was performed with whole-genome data of *Enterococcus* isolated from *Galleria mellonella* larvae, and as a result, *Enterococcus innesii* was finally identified. Relevant results have been corrected in Fig2C.

The related results were modified as follows in the text.

“Species classification was performed for Gram-positive bacteria, which are dominant in *G. mellonella* larvae. Whole genome analysis of bacteria isolated from larvae was performed through previous studies (ref), and when identified based on relevant whole genome data, it was finally identified as *Enterococcus innesii* (Fig. 2C).” P8 L133-137

In addition, the related reference papers below have been added.

“Chung J-H, Jeong H, Ryu C-M.2018. Complete Genome Sequences of *Enterobacter cancerogenus* CR-Eb1 and *Enterococcus* sp. Strain CR-Ec1, Isolated from the Larval Gut of the Greater Wax Moth, *Galleria mellonella*. *Genome Announc.* 15:e00044-18.”

10. Lines 227-229... All results even if supplemental should be discussed in the results, as we have no context for how and why this experiment was done.

Response: Thank you. We have added information related to every Figure data to the manuscript.

11. Line 251-255: Some network analyses of other statistical methodology could be presented to see the correlations between the *G. mellonella* gene expression with the microbiome at each stage. Also since this is a linear vent, one could do linear mixed model or time dependent models to look at the change in gene expression, and change in microbiome overtime to see how they are correlated.

Response: Thanks for the advice. As advised, correlation analysis between microbiome and RNAseq data was performed through Sparse canonical correlation analysis, and the results are added in Fig S2. Relatedly, results and methods have also been added to the text.

Result: “In the case of microtubule (GO0005874) and DNA recombination (GO0006310), it was confirmed that they increased significantly in pupae in the results of Sparse Canonical Correlation analysis (Fig. S2).”

Method: “Sparse Canonical Correlation analysis. Sparse canonical correlation analysis (CCA) was performed to analyze the correlation between microbiome data and RNAseq expression in the larval and pupal stages of *Galleria mellonella*. The microbiome database used the results obtained from the qiime2 analysis, and after removing the taxon with a read number of 0 in the two treatment groups, the taxon with a difference of more than two times was used. In the case of RNAseq data, the DEG analysis results were used, genes with 0 in both treatment groups were removed, and a gene list with a difference of more than 2 times between treatment groups was performed. CCA analysis was performed using the iSFun package of the R program. In the obtained data, the gene list was grouped by performing Gene Ontology analysis using Panther GO-Slim. The results were displayed as a heat map through the R program.” P21 L421-430

12. Line 328...with the respective/target AMP?? You tested more than one, yes?

Response: Yes, we tested more than one AMP from larvae and pupae to test their effects on the growth of three major gut bacteria. Therefore, the contents of the text have been amended.

“To investigate the growth inhibition effect of extracted crude AMPs on intestinal bacteria from *G. mellonella*, bacteria were adjusted to OD600 = 0.1 and treated with AMP at 1/100 dilution, respectively.” P22 L436-438

13. Main paper says LC/MS..methods says LC-ESI-QTOF-MS/MS, these are two different methods, please clarify

Response: Sorry for making confusing. We changed all LC-MS to LC-ESI-QTOF-MS in the text.

14. Table 1: Have other papers in *G. mellonella* reported such low #s for OTUs in larval stage??

Response: Yes, we found the relevant references as below:

“Allonsius et al., 2019 The microbiome of the invertebrate model host *Galleria mellonella* is dominated by *Enterococcus*.”

As a result of recent microbiome analysis of each part of *G. mellonella*, it was confirmed that *Enterococcus* is mainly present in the intestine.

15. Table 1: Probably best to define 3/4-YBK in table footnote

Response: Thank you We have intuitively modified the Table 1 annotations to make them easier to understand.

16. Figure 2: Please label graphs appropriately in panel A. The 16S reads only map to the genera level, not the species level

Response: We modified the label in Fig 2 to the genus level.

17. Figure 2 B, to limit confusion, I would color *Enterobacter* and *Pseudomonas* differently. They are very different bacteria and should not be lumped together.

Response: We modified Figure 2 to fit the data.

18. What statistical test is used in Figure 3?? State in methods or figure legend

Response: We analyzed by Two-way repeated measures ANOVA with Bonferroni post-hoc test using R program. Added related information to method.

“Statistical analysis. The results were obtained by performing one-way repeated measures ANOVA using the Bonferroni post-test and two-tailed student's t test by the R program.” P23 L456-457

[Reviewer 2]

This work by Kong et. al., seeks to evaluate whether the midgut microbiome of the *G. mellonella* wax moth controls its metamorphosis. They show that the bacteria species *Enterococcus gallinarum* is uniquely associated with larval stages of *G. mellonella*. When the bacteria are depleted using antibiotics there is a rapid onset of pupation, and this rapid pupation is rescued by the reintroduction of *E. gallinarum*. The authors claim that the onset of pupation via hormone production induces the expression of antimicrobial peptides in response to the *E. gallinarum* present in the gut, which controls the level of *E. gallinarum* in the midgut, which then results in onset of pupation due to bacterial depletion.

The story presented by the authors is very interesting and could provide important insight into how bacteria regulate or are intimately involved in the *G. mellonella* lifecycle. Further, the model presented is insightful, and illustrated beautifully in Figure 6. However, I have some hesitations regarding the conclusions drawn from the data presented, mainly due to it seemingly being more correlative than causative, while claiming causation.

Response: Thanks for your positive feedback. As you said, to analyze the relationship between gut microbes and pupalization in the development of insects, we presented the possibility of relevance based on the results of RNA-seq analysis of the gut microbiome and the host, suggesting that these results are correlated. Therefore, to confirm a clear causal relationship between them, more in-depth research should be conducted based on the results of this study in the future. However, the correlation between the microbial community in the insect gut and pupalization, which is most dramatic in the development of insects, is presented for the first time in this study, which is considered to be a very meaningful result in terms of evolution or development of insects.

Figures 1 through 3 are important for establishing the midgut microbiome that is present during different life stages and showing that depletion and re-addition of specific members of the microbiome affects the time until pupation. The authors convincingly show that treatment with antibiotics, and then re-introducing bacteria into the midgut speeds up and restores the rate of pupation, respectively. Figures 4 and 5 show what genes and antimicrobial peptides are expressed during different life stages, which correlates to changes in the midgut microbiome. While they show that purified AMPs from the larval stages selectively kill the *E. gallinarum*, they do not show a causative link or directionality between microbiome composition and expression of antimicrobial peptides, as they suggested through a feedback loop mechanism.

Method: “

This can be accomplished by looking at gene expression or AMP expression following antibiotic treatment and/or re-introduction of the microbiome. They do not show that the AMP/hormone expression is changed when the *E. gallinarum* is absent, which seems to be a major feature of their feedback loop mechanism.

Response: Thanks for the advice. We further confirmed the expression of AMP-related genes in *G mellonella* by the treatment of *Enterococcus*.

To this end, *E. innesii* isolated from the intestine were fed after vancomycin treatment, and the expression of gallerimycin, cecropin, IMPI, and apolipophorin-related genes was compared over time. As a result, it was confirmed that AMP-related genes temporarily increased by vancomycin treatment, but recovered quickly, and in the case of *E. innesii*, gene expression was induced and maintained. Interestingly, in the case of Apolipophorin, which increased protein production in pupae, it was confirmed that gene expression increased only by treatment with *E. innesii*. These results suggest an association between *E. innesii* and AMPs, particularly apolipophorin. We added these results to Fig. 7 and explained the related contents in the manuscript.

“Changes in AMP-related gene expression through *Enterococcus* fed. As a result of the above, we confirmed that the pupalization of *G. mellonella* changes according to the presence or absence of *Enterococcus*. In addition, it was confirmed that only *Enterococcus* showed inhibition of growth when AMP of pupae was extracted and treated with the isolated bacteria. Therefore, in order to identify *E. innesii*-specific AMP,

the expression of AMP-related genes was assayed in *G. mellonella* treated with vancomycin and then fed with *E. innesii*. As a result, it was confirmed that the expression of Gallerimycin, Cecropin, and IMPI related gene rapidly increased and then disappeared after 24 hours of treatment with Vancomycin. In the case of feeding with *E. innesii*, unlike Vancomycin treatment, it was confirmed that gene expression was maintained continuously even on the 48 hours after feeding with *E. innesii*. On the other hand, in the case of Apolipphorin, gene expression was not observed when treated with vancomycin, but interestingly, when *E. innesii* was fed, it was confirmed that gene expression was specifically increased (Fig. 7)." P13 L248-259

Response: To prevent misunderstanding, in Figure 6, we mark the parts (hormone and AMP) where the causal relationship has not been clearly proven as a result of our research with "?".

I think the conclusion that *E. gallinarum* exclusively affects pupation rate is not completely proven. While re-addition of *E. gallinarum* alone in the midgut of the larvae convincingly restores normal pupation rate following antibiotic treatment, it is not clear whether the effect is due specifically to the presence of *E. gallinarum* or due to the restored levels of any *Enterococcus* spp./bacteria present in the midgut. It is unclear whether re-introduction of large quantities of other *Enterococcus* species or an unrelated gram-positive bacterium would have similar effects.

Response: Thanks for your valuable comment. We conducted additional experiments to confirm this. In the additional experiment, isolated *E. gallinarum* and *E. mundtii* were fed after vancomycin treatment. The pupation rate was evaluated following feeding *Bacillus subtilis* 168, one of the other model Gram-Positive bacteria. As a result, in the case of *E. gallinarum* and *E. mundtii*, which are *Enterococcus* sp. present in the intestine, it was confirmed that the change in pupation by antibiotic treatment was recovered to the same level as the untreated group. On the other hand, in the case of other *B. subtilis* 168, it was tested that the recovery of pupation did not appear. These results indicate that *Enterococcus* present in the intestine is

important for pupation, and external Gram-Positive bacteria do not play a role in pupation. We have added these results to Fig. 3E and F.

“Furthermore, the reaction of other Gram-positive bacteria in the pupation recovery of *Galleria mellonella* was assayed. In other *Enterococcus mundtii* (Em) detected in *G. mellonella* intestinal microbes, it was confirmed that pupation was recovered similarly to that of the PBS-treated group. However, in *Bacillus subtilis* 168 (Bs) treatment, which was not observed in the intestine, pupation was similar to vancomycin alone. On the other hand, in the case of Em, pupation was recovered to a level similar to that of the antibiotic-free control group as in Ei (Fig. 3E and 3F).” P9 L159-165

Additionally, the authors did not confirm whether the midgut microbiome was depleted by the addition of vancomycin. While it would make intuitive sense, it would be good to confirm, in case the drug is not as effective or there are persistent drug-resistant *Enterococcus* spp still present.

Response: Thanks for the advice. We took the advice and conducted further experiments using culture techniques to detect the reduction of gut microbiota after antibiotic treatment. First, for the culture method, we secured Rifampicin-resistant *E. gallinarum* and *E. mundtii*, fed them into *Galleria mellonella*, treated with antibiotics, and confirmed the survival of each strain in the intestine by culturing them in a medium containing Rif. As a result, after 24 hours, bacteria of 10^5 to 10^6 survived in the intestines not treated with antibiotics, while bacteria treated with antibiotics showed detection of less than 10, confirming that *Enterococcus* was depleted by antibiotic treatment. These results have been added to Fig. S4.

“After vancomycin treatment, the survival of antibiotic-induced Gram positive bacteria including *Enterococcus* spp. in the intestine was evaluated by culturing the isolated *E. innesii* and *E. mundtii* by marking them for resistance to rifampicin antibiotic (referred to as EgrifR and EmrifR respectively). As a result, it was confirmed that each strain was present in the intestine at $10^5 - 10^6$ cfu/g 24 hours after feeding. However when treated with vancomycin, both strains showed less than 10-100 cfu/g (Fig. S4) indicating the effectiveness of vancomycin.” P141-147

I found the Materials and methods section to be lacking information on some of the experiments performed, namely:

- Rearing conditions of the larvae that were purchased, and what they were fed once they hatched from the eggs.

Response: Thanks for the advice. We added the following feed composition and conditions for breeding *Galleria mellonella*.

“Each larva was reared in the dark in a 6.5 L plastic container (31.5 X 23 X 11.5 cm, Lock&Lock Co.Ltd., S. Korea) filled with nutrient-rich food at 30 °C. The food consisted of wheat bran 300 g, rice bran 300 g, yeast extract 2.25 g, CaCO₃ 1 g, glycerol 125 mL, honey 87.5 mL, and vitamin B complex 300 mg. Feed was replaced once a week.” P18 L343-347

- Definition of TSA and TSB media.

Response: Thank you. We have added a description of TSA and TSB. P15 L280, L285

- How egg and adult microbiomes were collected.

Response: Thanks for the advice. We have added methods for obtaining microbiomes from eggs and pupae.

“At each stage, *G. mellonella* populations were collected and rinsed with 100% ethanol and sterile water. For eggs, DNA isolation was performed after surface sterilization with 100% ethanol, and for larvae and pupae, gut samples were collected for subsequent analysis.” P19 L177-379

- Definition of how instars were differentiated.

Response: Thanks for the advice. We classified the instar of the larvae through the size of the larvae, and the related literature is as follows.

Reference: Wojda et al., 2020. The greater wax moth *Galleria mellonella*: biology and use in immune studies. *Pathog Dis.* 2020 Dec; 78(9).

- Were the AMP activity experiments done on whole *Galleria* protein extract? The section beginning at Line 322 does not describe purification of the AMP (although the section beginning on line 332 does somewhat describe that purification process). Was the same purification of small peptides via a gel performed, or were bacteria just treated with all the protein from homogenized *G. mellonella*?

Response: Since this experiment aims to confirm the change of AMP according to the developmental stage of *G. mellonella* and the subsequent effect on bacteria, crude AMPs extracted from the larval and pupa stages were extracted, and the effect on the growth of the isolated bacteria investigated. Added treatment for germs to the text.

“To investigate the growth inhibition effect of extracted crude AMPs on intestinal bacteria from *G. mellonella*, bacteria were adjusted to $OD_{600} = 0.1$ and treated with AMP at 1/100 dilution, respectively.” P22 L436-438

Specific comments:

Abstract: I found this section to be a little too dry and written in passive voice.

Response: Thanks for the advice. We have modified the Abstract as follows for easier access.

“Microbes found in the digestive tracts of insects are known to play an important role in their host’s behavior. Although Lepidoptera is one of the most varied insect orders, the link between microbial symbiosis and host development is still poorly studied. In particular, little has been clarified about the relationship between gut bacteria and metamorphosis. Therefore,

we explored gut microbial biodiversity throughout the life cycle of *G. mellonella* using pyrosequencing with the V1 - V3 regions. As a result, *Enterococcus* spp. was abundant and active in larvae, and *Enterobacter* spp. was higher in pupae. Interestingly, eradication of *Enterococcus* spp. from the digestive system accelerated the larval-to-pupal transition. Furthermore, genetic changes in the host through transcriptome analysis showed that immune response genes were up-regulated in pupae, whereas hormone genes were up-regulated in larvae. In particular, host-produced antimicrobial peptides changed at various developmental stages, and some of these antimicrobial peptides were shown to inhibit the growth of *Enterococcus gallinarum*, a dominant bacterial species in the gut of *G. mellonella* larvae. This study suggests the importance of the gut microbiome in *G. mellonella* metamorphosis and elucidates the effect of gut microbiome modification on host metamorphosis.” P2 L21-34

Line 37: the directionality discussed in this sentence is not well explained

Response: Thanks for the advice. We reduced the part about directionality, and modified the contents as follows for more accurate information.

“Here, we firstly demonstrated that the presence of *Enterococcus* spp. provides a driving force on an insect metamorphosis. Study on RNA-sequencing and peptide production revealed that antimicrobial peptides in *Galleria mellonella* gut did not inhibit *Enterobacteria* spp. but *Enterococcus* spp. in fourth instar resulting to accelerating pupation.” P3 L28-41

Introduction: It might be good to start off with an insect microbiome introduction paragraph.

Response: Thank you. We modified the sentence at the beginning of the introduction and started the overall opening paragraph with the relationship between insects and the microbiome. Here

are the fixes:

“The study of microbiome composition and its interactions in insects has become more precise and diverse with the development of next-generation sequencing (NGS) technologies.” P4 L44-45

Line 68: Change "adipose" to "fat"

Response: Thanks for the advice. We changed “adipose” to “fat” as advised. P5 L67

Do you think the microbial composition of the eggs is related to the microbes from the environment (container, food) or from the reproductive tract of the female moth?

Response: Since this study was mainly focused on the relationship between pupation of *G. mellonella* and intestinal microorganisms, it is difficult to pinpoint the cause of the composition of microorganisms in eggs. However, the results of the microbial community in *G. mellonella* were confirmed similar to the results of other studies.

Reference: “Gohl P. et al. 2022. Diet and ontogeny drastically alter the larval microbiome of the invertebrate model *Galleria mellonella*. Canadian Journal of Microbiology.”

Figure 1: Define the instar abbreviations.

Response: Thanks for the advice. We added explanations for 3-YBK and 4-YBK indicated in the figure legends.

Line 101 to 104: Not sure this is a logical conclusion from the data in Figure 1, seems more like the conclusion from the subsequent figures.

Response: Thanks for the advice. We agree with your advice and have modified the relevant content as follows.

“These results suggest that the intestinal microbiome of *G. mellonella* differs according to development, and in particular, *Enterococcus* spp., which accounts for more than 90% of the larval intestinal microbiome during pupation, is converted to *Enteobacter* spp. in the pupal stage.” P6 L105-107

Figure 4: It is unclear what the expression levels are being compared to. Fold change compared to what conditions/life stage? Pupae to larvae, larvae to pupae? What about when both are shown? There might be a better way to display or explain what is being shown.

Response: Sorry. We performed PANTHER Go-slim analysis based on the DEG analysis results obtained from RNAseq and illustrated the fold enrichment values obtained as a result of the analysis. Therefore, we modified the Fold expression in Figure 5 to Fold enrichment.

Lines 245-246: I do not think these claims are supported, as there is only in vitro evidence that the AMPs kill *E. gallinarum*, not that they actually control *E. gallinarum* levels in the larval midgut.

Response: Thanks for the advice. We agree with the advice and amend the contents as follows.

“Interestingly, we confirmed that various AMPs are present throughout the developmental stages, and especially AMPs in fourth instars and pupae can affect the reduction of *Enterococcus innesii* at the pupal stage (Fig. 6).” P15 L300-302

Discussion: All data should be originally introduced in the results section, including the supplementary figures. Figure S3 is nice data and could be incorporated into one of the main figures. N

Response: Thank you. We modified the text to include all data, and Fig. S3 included in main figure 3B.

“When polymyxin B, an antibiotic that targets only Gram-negative bacteria, was treated, there was no change in pupa rate as similar as a PBS treatment (Fig. 3B).” P8 L148-150

Figure 6 is very impressive and well-illustrated!

Response: Thank you very much.

April 10, 2023

Dr. Choong-Min Ryu
Korea Research Institute of Bioscience and Biotechnology
Infectious Disease Research Center
125, Gwahak-ro, Yuseong-gu
Daejeon 34141
Korea (South), Republic of

Re: Spectrum02780-22R1 (Population dynamics of intestinal Enterococcus modulate Galleria mellonella metamorphosis)

Dear Dr. Choong-Min Ryu:

Thank you for submitting your manuscript to Microbiology Spectrum. As you will see your paper is very close to acceptance. Please modify the manuscript along the lines I have recommended. As these revisions are quite minor, I expect that you should be able to turn in the revised paper in less than 30 days, if not sooner. If your manuscript was reviewed, you will find the reviewers' comments below.

When submitting the revised version of your paper, please provide (1) point-by-point responses to the issues raised by the reviewers as file type "Response to Reviewers," not in your cover letter, and (2) a PDF file that indicates the changes from the original submission (by highlighting or underlining the changes) as file type "Marked Up Manuscript - For Review Only". Please use this link to submit your revised manuscript. Detailed instructions on submitting your revised paper are below.

Link Not Available

Sincerely,

Vittal Ponraj Ph.D SM(ASCP)CM

Editor's Comments:

As the Reviewers have pointed out below, the modified version has introduced innumerable typos and grammatical errors that requires Professional proof-reading. Just to point out some glaring comprehensive errors/typos in the Abstract section (without citing similar errors/typos throughout the manuscript):

Lines 26-28- The sentence needs to be rephrased/edited.

For eg: "Enterobacter spp. swifited to major in pupae."-

"----led to be accelerated the larval-to-pupal transition."

Not clear what the authors are trying to imply/convey.

Line 40- "RNA-sequencing"- Typo. "RNA-sequencing"

I highly recommend using a Professional Language Editing Services before resubmission. I am providing the link listed on the ASM website for your reference <https://journals.asm.org/language-editing-services>

Disclaimer: ASM has no connection or any reciprocal financial arrangements with any of these companies; the list below is provided for your convenience. These services are completely independent of ASM and of ASM's peer review and editing process, and the use of such services has no bearing on the acceptance of a manuscript for publication.

Reviewer comments:

Reviewer #1 (Comments for the Author):

- I'm not sure what is trying to be said in Line 39-40 of edited version "antimicrobial peptides in *Galleria mellonella* gut did not inhibit *Enterobacteria* spp. but *Enterococcus* spp. in fourth instar resulting to accelerating pupation" ..this should probably be made clear in more "lay" terms for the readership
- Italicize all genus and species names
- In much of the edited content, the English grammar and spelling are not correct and sometimes difficult to understand. Example: "Changes in AMP-related gene expression through *Enterococcus* fed" *Enterococcus* fed what? *Enterococcus* feeding?
- There are places where vancomycin is capitalized, and others where it is not (it should not)
- Lines 277-280...I am not following what this is trying to say the way this worded
- many typos in many places
- Fig 1 A- Is this an average abundance? I'm sure you did replicates to represent each stage. Please be clear in figure legend how many replicates are represented and how averaged.
- Fig 2 A. Similar to above, this has to be avg. reads per group, please either represent box plot with points with mean/std dev, or violin plot
- Did you reperform all of these experiments with *Enterococcus innesii* or did you misname all of your experiments as *E. gallinarium* in the previous version but then find it was actually *E. innesii*?

Reviewer #2 (Comments for the Author):

The changes made by the authors have majorly improved the paper, and strengthened their findings. I have a few suggestions/comments below:

I still think a better way to phrase Page 15, line 300-302 would be "...can affect the reduction of *Enterococcus innesii* in vitro" because the authors are showing that the AMPs killed the bacteria or slowed the growth in vitro rather than during infections. Presumably, it would also kill the bacteria in vivo but just for the sake of clarity.

Lines 234-243 and 430-440 - If I am understanding correctly, it is a purification of whole protein from the larva, so this includes all the proteins present, not just AMPs. While AMPs are present and active, describing it as a "purification of AMP" or "crude AMPs" doesn't seem the most accurate description given many other proteins present. Maybe calling it the "antimicrobial growth inhibition of crude protein extract from the different stages of development" or phrasing it along those lines.

The Figure 7 legend should describe what the arrow indicates and should reference the individual panels in the figure.

I think "pupation" would be a better word choice than "pupalization" (line 166, 245, and 582), or pupariation (lines 150, 567, 584, 586).

Vancomycin is spelled wrong in Figure 4B,C.

In figure 4, it is difficult to tell whether the "PBS treatment" group in B and C are referring to initial treatment with PBS instead of vancomycin on Day 0 (i.e. the larvae are not bacteria-depleted) or if it is referring to the second treatment on Day 2 as a mock re-introduction of bacteria or bacterial components). Really nice experiment though to show where the cell components responsible for this induction of pupation are.

Thank you for including the *B. subtilis* experiment and the experiments showing the changes to AMP following exposure to bacteria.

Line 33 - "microbiota" spelled incorrectly.

Line 316 - I think describing it as "co-evolution" might need further clarification or re-phrasing.

Preparing Revision Guidelines

- Point-by-point responses to the issues raised by the reviewers in a file named "Response to Reviewers," NOT IN YOUR

COVER LETTER.

- Upload a compare copy of the manuscript (without figures) as a "Marked-Up Manuscript" file.
- Each figure must be uploaded as a separate file, and any multipanel figures must be assembled into one file.
- Manuscript: A .DOC version of the revised manuscript
- Figures: Editable, high-resolution, individual figure files are required at revision, TIFF or EPS files are preferred

Please return the manuscript within 60 days; if you cannot complete the modification within this time period, please contact me. If you do not wish to modify the manuscript and prefer to submit it to another journal, please notify me of your decision immediately so that the manuscript may be formally withdrawn from consideration by Microbiology Spectrum.

Point by Point Response

[Editor's comments]

Comment: As the Reviewers have pointed out below, the modified version has introduced innumerable typos and grammatical errors that requires Professional proof-reading. Just to point out some glaring comprehensive errors/typos in the Abstract section (without citing similar errors/typos throughout the manuscript):

Response: Thank you for your comments, as you and reviewers' suggestions, we got professional English proof-reading service for our manuscript. We've attached the invoice at the last page of this response.

Comment: Lines 26-28- The sentence needs to be rephrased/edited. For eg: "Enterobacter spp. swiftd to major in pupae."-"-----led to be accelerated the larval-to-pupal transition." Not clear what the authors are trying to imply/convey.

Response: As your suggestion, we modified the sentences to "*Enterococcus* spp. were abundant in larvae while *Enterobacter* spp. were predominant in pupae. Interestingly, eradication of *Enterococcus* spp. from the digestive system accelerated the larval-to-pupal transition."

Comment: Line 40- "RNA-sequencing"- Typo. "RNA-sequencing"

Response: As your suggestion, we corrected the typo error.

Comment: I highly recommend using a Professional Language Editing Services before resubmission. I am providing the link listed on the ASM website for your reference <https://journals.asm.org/language-editing-services>

Response: Thank you for your comments, as you and reviewers' suggestions, we got professional English proof-reading service for our manuscript. We've attached the invoice to the last page of this response.

[Reviewer 1]

Thanks for your valuable suggestions and comments, we've tried to do our best to follow your suggestions. Based on your comments, our manuscript has been much improved!

Comment: I'm not sure what is trying to be said in Line 39-40 of edited version "antimicrobial peptides in *Galleria mellonella* gut did not inhibit *Enterobacteria* spp. but *Enterococcus* spp. in fourth instar resulting to accelerating pupation" ..this should probably be made clear in more "lay" terms for the readership

Response: Thanks for your suggestion. We modified this sentence using more lay terms. It changed to “RNA-sequencing and peptide production subsequently revealed that antimicrobial peptides targeted against microorganisms in the gut of *Galleria mellonella* (wax moth) did not affect *Enterobacteria*, but did kill *Enterococcus* leading to promote moth pupation in the transition from larva to pupa.”

Comment: Italicize all genus and species names.

Response: Thanks for your suggestion. We changed all genus and species in italic.

Comment: In much of the edited content, the english grammar and spelling are not correct and sometimes difficult to understand. Example: "Changes in AMP-related gene expression through *Enterococcus* fed" *Enterococcus* fed what? *Enterococcus* feeding?

Response: Thanks for your suggestion. We changed this section title to “Changes in AMP-related gene expression following *Enterococcus* inoculation”. In addition, we got professional English editing service for our manuscript. The invoice is attached at last page of this response.

Comment: There are places where vancomycin is capitalized, and others where it is not (it should not)

Response: Thanks for your suggestion. We unified all words to “vancomycin”.

Comment: Lines 277-280...I am not following what this is trying to say the way this worded

Response: Thanks for your comment. We modified the sentence to “In addition, the restoration of pupation solely in the presence of *Enterococcus*, but not other Gram-positive bacteria, provides evidence for the specificity of the response to *Enterococcus*”.

Comment: many typos in many places

Response: Thanks for your comment. To avoid typos in our manuscript we got professional English editing service for our manuscript. The invoice is attached on the last page of this response.

Comment: Fig 1 A- Is this an average abundance? I'm sure you did replicates to represent each stage. Please be clear in figure legend how many replicates are represented and how averaged.

Response: We used 8 samples (replications) for each stage for the sequencing. We noticed it in Figure legends as you request.

Comment: Fig 2 A. Similar to above, this has to be avg. reads per group, please either represent box plot with points with mean/std dev, or violin plot

Response: We used 8 samples (replications) for the sequencing but visualized it as pooling data at the first time. However, as you request, to visualize this as box plot, we sequenced more samples during the 2nd revision and visualized it again as box plot. Hence total 12 samples for each group were used.

Comment: Did you reformed all of these experiments with *Enterococcus innesii* or did you misname all of your experiments as *E. gallinarium* in the previous version but then find it was actually *E. innesii*?

Response: Thanks for your comments. As you guess, we did not change the bacterial strains used in this study. Although, at the beginning of this work, we classified the bacterial strains as *E. gallinarium* based on 16S rRNA sequencing, however, during the revision steps, we performed whole genome sequencing for bacterial strain and newly classified it to *E. innesii*.

[Reviewer 2]

Thanks for your valuable suggestions and comments, we've tried to do our best to follow your suggestions. Based on your comments, our manuscript has been much improved!

Comment: The changes made by the authors have majorly improved the paper, and strengthened their findings. I have a few suggestions/comments below:

Thanks for your valuable suggestions and comments, we've tried to do our best to follow your suggestions. Based on your comments, our manuscript has been much improved!

Comment: I still think a better way to phrase Page 15, line 300-302 would be "...can affect the reduction of *Enterococcus innesii* in vitro" because the authors are showing that the AMPs killed the bacteria or slowed the growth in vitro rather than during infections. Presumably, it would also kill the bacteria in vivo but just for the sake of clarity.

Response: Thanks for your comment. As your suggestion, we used "in vitro" in this sentence. So, final version is "Especially AMPs in fourth instars and pupae can affect the reduction of *Enterococcus innesii* in vitro"

Comment: Lines 234-243 and 430-440 - If I am understanding correctly, it is a purification of whole protein from the larva, so this includes all the proteins present, not just AMPs. While AMPs are present and active, describing it as a "purification of AMP" or "crude AMPs" doesn't seem the most accurate description given many other proteins present. Maybe calling it the "antimicrobial growth inhibition of crude protein extract from the different stages of development" or phrasing it along those lines.

Response: Thanks for your comment. We used term “crude peptides extracts including AMPs” in both sentences.

Comment: The Figure 7 legend should describe what the arrow indicates and should reference the individual panels in the figure.

Response: Thanks for your comment. For better understanding, we indicated the arrow on the figure legend and individual panels in the figure using alphabet.

Comment: I think "pupation" would be a better word choice than "pupalization" (line 166, 245, and 582), or pupariation (lines 150, 567, 584, 586).

Response: Thanks for your comment. We used term “pupation” for all sentences.

Comment: Vancomycin is spelled wrong in Figure 4B,C.

Response: Thanks for your comment. We corrected.

Comment: In figure 4, it is difficult to tell whether the "PBS treatment" group in B and C are referring to initial treatment with PBS instead of vancomycin on Day 0 (i.e. the larvae are not bacteria-depleted) or if it is referring to the second treatment on Day 2 as a mock re-introduction of bacteria or bacterial components). Really nice experiment though to show where the cell components responsible for this induction of pupation are.

Response: Thanks for your comment! Sorry we checked the figure and agreed that it is hard to understand. Hence, we slightly changed the figure 4A for better understanding. ‘Vancomycin’ indicates PBS treatment following Vancomycin treatment. ‘PBS’ indicates PBS treatment after PBS treatment.

Comment: Thank you for including the *B. subtilis* experiment and the experiments showing the changes to AMP following exposure to bacteria.

Response: We thank you!

Comment: Line 33 - "microbiota" spelled incorrectly.

Response: Thanks for your comment. We corrected the spell.

Comment: Line 316 - I think describing it as "co-evolution" might need further clarification or re-phrasing.

Response: Thanks for your comment. We rephrased the sentence to “This data suggests that specific interaction between the Gram-positive bacteria present in the intestine and host metamorphosis regulation has occurred.”

Company no 04150179
VAT No 899613067

Invoice

Invoice number: 19715
Invoice date: 25/04/2023
Choong-Min Ryu
125 Kwahangro Youseong,
Daejeon,
South Korea,
305-806

cmryu@kribb.re.kr

Job No.	Title	Services	Rate	No. of Words	Total
19715	Population dynamics of intestinal Enterococcus modulates Galleria mellonella metamorphosis	Editing Service: Standard Speed: Regular	USD 0.1100 per word	2051	\$225.61

Sub-total in USD: \$225.61

Total Charge \$225.61

May 24, 2023

Dr. Choong-Min Ryu
Korea Research Institute of Bioscience and Biotechnology
Infectious Disease Research Center
125, Gwahak-ro, Yuseong-gu
Daejeon 34141
Korea (South), Republic of

Re: Spectrum02780-22R2 (Population dynamics of intestinal Enterococcus modulate *Galleria mellonella* metamorphosis)

Dear Dr. Choong-Min Ryu:

Your manuscript has been accepted, and I am forwarding it to the ASM Journals Department for publication. You will be notified when your proofs are ready to be viewed.

Sincerely,

Vittal Prakash Ponraj Ph.D., SM(ASCP)CM
Editor, Microbiology Spectrum
